# Hepatic metabolism of grazing cows of two Holstein strains under two feeding strategies with different levels of pasture inclusion

**Mercedes García-Roche**[1,2]*, **Daniel Talmón**[1], **Guillermo Cañibe**[1], **Ana Laura Astessiano**[1], **Alejandro Mendoza**[3], **Adriana Cassina**[2], **Celia Quijano**[2], **Mariana Carriquiry**[1]

**1** Facultad de Agronomía, Departamento de Producción Animal y Pasturas, Universidad de la República, Montevideo, Uruguay, **2** Facultad de Medicina, Centro de Investigaciones Biomédicas (CEINBIO) and Departamento de Bioquímica, Universidad de la República, Montevideo, Uruguay, **3** Instituto Nacional de Investigación Agropecuaria, Programa Nacional de Producción de Leche, Ruta, Semillero, Uruguay

* mercedesg@fagro.edu.uy

## Abstract

The objective of the study was to characterize adaptations of hepatic metabolism of dairy cows of two Holstein strains with varying proportions of grazing in the feeding strategy. Multiparous autumn calving Holstein cows of New Zealand (NZH) and North American (NAH) strains were assigned to a randomized complete block design with a 2 x 2 factorial arrangement with two feeding strategies that varied in the proportions of pasture and supplementation: maximum pasture and supplementation with a pelleted concentrate (MaxP) or fixed pasture and supplementation with a total mixed ration (FixP) from May through November of 2018. Hepatic biopsies were taken at - 45 ± 17, 21 ± 7, 100 ± 23 and 180 ± 23 days in milk (DIM), representing prepartum, early lactation, early mid-lactation and late mid-lactation. The effects of DIM, feeding strategy (FS), strain and their interactions were analyzed with mixed models using repeated measures. Cows of both strains had similar triglyceride levels, mitochondrial function and carnitine palmitoyltransferase activity in liver during lactation. However, there was an effect of DIM and FS as liver triglyceride was higher for the MaxP strategy at 21 DIM and both mitochondrial function and carnitine palmitoyltransferase activity in liver were lower for the MaxP strategy at 21 DIM. Hepatic mitochondrial function and acetylation levels were affected by the interaction between strain and feeding strategy as both variables were higher for NAH cows in the MaxP strategy. Mid-lactation hepatic gene expression of enzymes related to fatty acid metabolism and nuclear receptors was higher for NZH than NAH cows. This work confirms the association between liver triglyceride, decreased hepatic mitochondrial function and greater mitochondrial acetylation levels in cows with a higher inclusion of pasture and suggests differential adaptive mechanisms between NAH and NZH cows to strategies with varying proportions of grazing in the feeding strategy.

**Data Availability Statement:** All relevant data are within the paper and its Supporting Information files.

**Funding:** M. García-Roche was supported by Comisión Académica de Posgrados – Universidad de la República fellowship BDDX_2018_1#49004502. D. Talmón was supported by Agencia Nacional de Investigación e Innovación fellowship POS_NAC_2017_1_141266. A. Cassina and C. Quijano were funded by grants of the Espacio Interdisciplinario – Centros, Universidad de la República 2015 and Comisión Sectorial de Investigación Científica grupos I+D 2022 (22620220100012UD). A. Cassina was also supported by the grant Comisión Sectorial de Investigación Científica grupos I+D 2014 (767). The project was funded by Comisión Sectorial de Investigación Científica of the Universidad de la República CSIC I+D 2018 ID 103 to M. Carriquiry and C. Quijano as well as by Agencia Nacional de Investigación e Innovación INNOVAGRO 2018: FSA_1_2018_1_152220 to M. Carriquiry and A. Cassina. A. Mendoza received funding from the project PL_21_0_00 of Instituto Nacional de Investigación Agropecuaria. The funders had no role in study design, data collection and analysis, decision to publish, or preparation of the manuscript.

**Competing interests:** The authors have declared that no competing interests exist.

# Introduction

High-yielding dairy cows undergo extreme metabolic adaptions during lactation. The transition from gestating to lactating represents a period of excessive nutrient demands which requires the coordination of multiple tissues and will be crucial for the success of the oncoming lactation [1,2]. Particularly, during the transition period, dairy cows are susceptible to metabolic disorders which may be caused in part by the excessive mobilization of lipid, protein, vitamin and mineral body reserves [3].

The inclusion of pastures in dairy systems represents lower feeding costs positioning pasture-based systems as a competitive alternative for the intensification of dairy systems, especially in exporting countries where controlling production costs is fundamental [4,5]. Furthermore, due to increasing demands from consumers for quality products produced sustainably–minimizing nitrogen leaching, soil compaction and greenhouse-gas emissions and maximizing animal welfare–pasture-based systems need to ensure sufficient dry matter intake and at the same time maintain a high-quality pasture [6]. When nutrient concentration in the diet is insufficient, excessive mobilization of body reserves during early lactation may result in high levels of negative energy balance (NEB) markers such as concentrations of non-esterified fatty acids (NEFAs), beta-hydroxybutyrate and liver triglyceride [7,8]. Particularly, high plasma NEFAs are associated with an increased inflammatory response, polyunsaturated fatty acids can bind to a family of nuclear receptors called peroxisome proliferator-activated receptors (PPARs) and downregulate inflammatory reaction [9]. The link between metabolic disorders and inflammatory-based diseases is often hypothesized to be oxidative stress [10] which could be mediated by increased reactive oxygen species (ROS) derived from the mitochondrial electron transport chain, endogenous ROS production from NADPH oxidases by immune cells or decreased antioxidant defenses [9].

The liver, plays a key role in the metabolism of carbohydrates, proteins and lipids [11] and is key in the regulation of cholesterol and triglyceride metabolism alongside the maintenance of homeostasis and blood glucose levels [12]. However, hepatic mitochondrial function–key in energy metabolism–is decreased in grazing cows during early lactation when compared to cows fed total mixed ration (TMR) and is associated with increased mitochondrial acetylation levels [13].

In line with this, the suitability of the Holstein strain is also a crucial matter, since the NZH strain has lower maintenance costs than the NAH strain during late lactation when managed in a grazing system with supplementation of a commercial concentrate [14]. Indeed, the NZH strain has shown to be more suitable for grazing systems than the NAH strain since it maintains higher average body condition score, body weight and energy and feeding efficiency [5,15,16].Alongside, from a metabolic standpoint, the NAH strain presents uncoupling of the somatotropic axis during early lactation while the NZH strain maintains homeostatic levels of plasma growth hormone and insulin like growth factor-I [17]. Also, a recent publication from our group showed that cows of the NZH strain had increased mitochondrial function when compared to cows of the NAH strain in a pasture-based system during late mid-lactation and this was explained by an upregulation of the gluconeogenic pathway in cows of the NAH strain, since several metabolites are pulled toward gluconeogenesis instead of being oxidized to $CO_2$ [18]. Herein, we hypothesize that NZH cows will present improved mitochondrial function, lower levels of mitochondrial acetylation and oxidative metabolism in liver than NAH cows only in the feeding strategy with higher proportions of grazed pasture.

## Materials and methods

The experiment was conducted at the Experimental Station "La Estanzuela" of the Instituto Nacional de Investigación Agropecuaria (Colonia, Uruguay) during 2018. Animal use and procedures were approved by the Animal Experimentation Committee (CEUA) of the Instituto Nacional de Investigación Agropecuaria, Uruguay (file number: INIA2017.2).

The experiment formed part of a larger study designed to evaluate the effect of cow strain (NZH vs. NAH) under different FS on individual animal and whole-farm biophysical performance in a randomized complete block design with a 2 x 2 factorial arrangement of treatments. A more detailed description of Holstein strains, animal and grazing management, as well as milk production, body weight (BW) and body condition score (BCS) from June 2017 through May 2019 was previously reported [5].

### Experimental design, animals and feed

Autumn calving multiparous Holstein cows of NZH (538 ± 63 kg BW and 3.23 ± 0.19 BCS at calving; N = 24) and NAH (582 ± 59 kg BW and 3.03 ± 0.28 BCS at calving; N = 24) strains were used. The objective of the study was to evaluate the effect of Holstein strain and feeding strategy on hepatic metabolism from pre-partum to late mid-lactation.

Both NZH and NAH strains had at least 75% of each cow's ancestors (2 generations: father and maternal grandfather) from New Zealand, or from the United States or Canada respectively [5]. In the present study, the NZH strain represented a progeny of eleven sires and the NAH strain represented a progeny of twelve sires. The NZH and NAH strains were selected based on their genetic records in the national genetic evaluation system (Mejoramiento y Control Lechero Uruguayo; https://www.mu.org.uy). Cows of each strain were selected according to the national economic-productive breeding value which includes milk production, fat, protein, udder health and fertility and the average milk protein and fat progeny difference [19,20]. The national economic-productive breeding value was 125 ± 10 and 106 ± 14 on average for NZH and NAH cows, respectively. The expected progeny difference was -141 ± 179 kg, +-0.13 ± 0.15% and +0.14 ± 0.07% for milk yield, milk fat and milk protein content for NZH cows, while it was +45 ± 177 kg, +0.06 ± 0.16% and +0.01 ± 0.07% for milk yield, milk fat and milk protein content for NAH cows.

In the prepartum period (-45 DIM) cows were managed as single groups within each strain, cows grazed on *Dactylis glomerata* and *Medicago sativa* mixed pasture and were offered a TMR (Tables 1 and 2) with water *ad libitum*. Cows were blocked in each strain group according to their expected calving date (May 6, 2018 ± 20 days) and lactation number (3.1 ± 1.0 lactations). At calving, cows were allocated in two types of FS, with varying proportions of grazed pasture: fixed (FixP) and maximum (MaxP) pasture. Cows grazed on daily strips of *Dactylis glomerata* and *Medicago sativa* mixed pasture or *Festuca arundinacea* pastures in a rotational-grazing manner. Each strain group grazed on separate paddocks to assure similar pasture allowance (5 cm above ground level) relative to their BW and to ensure strains behaved independently and avoid dominance. Water was offered *ad libitum* in each paddock or feedpad. Details of estimated DM and nutrient intake, average herbage mass, chemical composition and metabolizable energy concentration of feedstuffs are presented in Tables 1 and 2, respectively. Variation in estimated DM intake (Table 1) is due to the fact that diets were formulated weekly.

Pasture allowance of FixP was fixed to 33% of the total annual DM intake–average annual pasture allowance accounting for seasonal variations–while the MaxP strategy had a flexible pasture allowance based on weekly pasture growth rate (PGR) in order to maximize grazed pasture. Grazing management guidelines have been detailed in Stirling et al., (2021), briefly,

**Table 1. Estimated dry matter and nutrient intake and average pre-grazing herbage mass value (mean ± SD) of grazing Holstein cows of North American and New Zealand strains in two feeding strategies with varying proportions of grazed pasture during lactation.**

| | Stage of lactation | | | | | | | | | | | | | |
| --- | --- | --- | --- | --- | --- | --- | --- | --- | --- | --- | --- | --- | --- | --- |
| | -45 DIM | | 21 DIM | | | | 100 DIM | | | | 180 DIM | | | |
| | NZH | NAH | NZH | | NAH | | NZH | | NAH | | NZH | | NAH | |
| | Prepartum diet | | FixP | MaxP | FixP | MaxP | FixP | MaxP | FixP | MaxP | FixP | MaxP | FixP | MaxP |
| Feedstuff (kgDM/cow/d) | | | | | | | | | | | | | | |
| Pasture | 7.8 ± 1.1 | 9.0 ± 0.9 | 5.8 ± 2.4 | 7.7 ± 3.6 | 7.3 ± 3.0 | 8.8 ± 4.2 | 6.1 ± 1.1 | 9.5 ± 4.7 | 7.7 ± 0.5 | 11.2 ± 4.8 | 6.7 ± 0.5 | 10.1 ± 1.9 | 7.7 ± 0.7 | 11.9 ± 2.4 |
| Concentrate | - | - | - | 6.7 ± 0.1 | - | 7.7 ± 0.2 | - | 7.3 ± 0.2 | - | 8.3 ± 0.3 | - | 6.3 ± 0.4 | - | 7.2 ± 0.4 |
| Conserved forage[1] | - | - | - | 4.5 ± 3.8 | - | 5.2 ± 4.0 | - | 4.5 ± 4.3 | - | 4.6 ± 4.4 | - | 4.1 ± 1.7 | - | 4.4 ± 0.8 |
| Total mixed ration[2] | 10.7 ± 1.1 | 13.1 ± 0.9 | 13.1 ± 2.6 | - | 14.8 ± 2.6 | - | 15.1 ± 0.9 | - | 16.5 ± 0.4 | - | 13.7 ± 0.8 | - | 15.8 ± 0.7 | - |
| Total DM intake | 18.5 ± 0.1 | 22.1 ± 0.1 | 18.9 ± 0.5 | 18.9 ± 0.5 | 22.0 ± 0.5 | 21.7 ± 0.6 | 21.2 ± 0.3 | 21.2 ± 0.3 | 24.2 ± 0.3 | 24.2 ± 0.3 | 20.5 ± 0.2 | 20.5 ± 0.1 | 23.5 ± 0.1 | 23.5 ± 0.1 |
| Average herbage mass (kgDM/ hectare) | 973 ± 460 | 1,175 ± 297 | 1,147 ± 659 | 1,105 ± 681 | 1,353 ± 809 | 1,604 ± 760 | 1,515 ± 449 | 1,482 ± 404 | 1,545 ± 495 | 1,471 ± 472 | 1,859 ± 519 | 1,507 ± 386 | 2,028 ± 534 | 1,573 ± 163 |
| Nutrient intake (kgDM/ cow/d) | | | | | | | | | | | | | | |
| CP | 2.9 ± 0.1 | 3.4 ± 0.05 | 3.6 ± 0.2 | 3.9 ± 0.5 | 4.1 ± 0.3 | 4.6 ± 0.3 | 4.0 ± 0.1 | 4.5 ± 0.6 | 4.5 ± 0.1 | 5.2 ± 0.6 | 3.7 ± 0.01 | 4.0 ± 0.8 | 3.9 ± 0.03 | 4.6 ± 0.9 |
| NDF | 8.0 ± 0.04 | 9.5 ± 0.01 | 7.3 ± 0.3 | 7.9 ± 0.2 | 8.8 ± 0.6 | 9.1 ± 0.2 | 8.3 ± 0.3 | 9.2 ± 0.6 | 9.1 ± 0.1 | 10.5 ± 0.6 | 8.8 ± 0.05 | 7.9 ± 1.4 | 10.3 ± 0.05 | 9.8 ± 1.7 |
| ADF | 5.0 ± 0.03 | 5.9 ± 0.01 | 4.4 ± 0.1 | 4.6 ± 0.3 | 5.3 ± 0.3 | 5.1 ± 0.2 | 5.1 ± 0.2 | 5.3 ± 0.3 | 5.2 ± 0.1 | 6.1 ± 0.4 | 5.2 ± 0.05 | 4.5 ± 0.8 | 6.3 ± 0.05 | 5.2 ± 0.9 |
| ME (MJ/cow/d) | 190 ± 1 | 227 ± 1 | 199 ± 7 | 170 ± 17 | 235 ± 4 | 200 ± 24 | 226 ± 2 | 223 ± 2 | 263 ± 3 | 254 ± 2 | 211 ± 3 | 180 ± 16 | 242 ± 1 | 210 ± 19 |

DM = Dry matter; DIM = Days in milk; NZH = New Zealand Holstein; NAH = North American Holstein; FixP = fixed pasture; MaxP = maximum pasture.

[1.] Composed of corn silage and pasture haylage in a 75:25 ratio in DM basis in average throughout the experimental period.

[2.] 65:35 forage to concentrate ratio DM basis during the prepartum period and 55:45 forage:concentrate ratio in DM basis during the experimental period. SD represents the variation within the experimental period.

weekly walks were performed and sward height was measured with a C-Dax Pasture Meter (C-Dax Systems Ltd.). Pasture allowance was calculated based on PGR and stocking rate, if PGR was not enough to cover nutrient requirements, conserved forage was offered. Pasture was allocated daily and pasture intake was estimated using pre and post grazing pasture heights with the C-Dax Pasture Meter. Predicted DM intake was estimated weekly according to the National Research Council (NRC) model for Dairy Cattle (2001). In the FixP strategy, as pasture DM intake was fixed to 33% of total DM intake, cows had access to one grazing session during the daytime after the AM milking session (0500 to 1400 h) and were supplemented with TMR in a feedpad after the PM milking session (1500 to 0400 h). In the MaxP strategy, cows had access to two grazing sessions after each milking session where they were offered an energy-protein concentrate.

Concentrate represented 33% of total DM intake as annual average for both FS. In FixP, concentrate was offered in the TMR and was a mixture of high-moisture corn grain (47% on a DM basis), soybean meal (37%), soy hulls or wheat bran (16%), urea, sodium bicarbonate, dicalcium phosphate, magnesium oxide, yeast, and a mineral-vitamin premix and in the MaxP strategy a pelleted commercial concentrate was offered in the milking parlor. Offered and refused supplements on the feedpad (TMR for FixP and conserved forage for MaxP) were weighed daily using the mixer's electronic scales, while offered and refused concentrate in the milking parlor (MaxP) was weighed every 14 d.

Cows were milked twice a day (0400 and 1500 h) and milk yield was recorded daily by a #7161-9005-062; Metatron P21, GEA Ltd. Milk samples were collected every 14 d. Solids corrected milk was calculated as: SCM (kg) = 12.3(F) + 6.56 (SNF)– 0.0752 (M); where F, SNF and M are expressed as kg of fat, solids-not-fat and milk, respectively [21]. Cow BCS (score 1 to 5; Edmonson et al., 1989) and BW were determined every 14 d.

**Table 2. Chemical composition and metabolizable energy concentration (means ± SD) of feedstuffs.**

| | -45 DIM | | 21 DIM | | | | 100 DIM | | | | 180 DIM | | | |
|---|---|---|---|---|---|---|---|---|---|---|---|---|---|---|
| | Pasture | TMR | Pasture | Concentrate | Conserved forage | TMR | Pasture | Concentrate | Conserved forage | TMR | Pasture | Concentrate | Conserved forage | TMR |
| DM (%) | 30.1 ± 5.5 | 54.3 ± 8.6 | 17.0 ± 5.5 | 90.1 ± 0.8 | 40.5 ± 7.9 | 50.9 ± 3.8 | 20.8 ± 4.1 | 88.8 ± 0.3 | 43.5 ± 10.0 | 54.3 ± 8.6 | 23.7 ± 2.1 | 88.7 ± 0.5 | 50.1 ± 13.0 | 54.3 ± 8.6 |
| CP (% DM) | 19.8 ± 2.3 | 12.8 ± 2.9 | 24.6 ± 1.9 | 20.6 ± 1.4 | 11.8 ± 4.0 | 16.2 ± 2.5 | 24.4 ± 1.6 | 22.4 ± 0.6 | 13.2 ± 4.6 | 16.3 ± 1.3 | 24.8 ± 2.8 | 21.6 ± 0.4 | 12.7 ± 5.5 | 14.7 ± 0.2 |
| NDF (% DM) | 45.2 ± 5.2 | 41.6 ± 3.6 | 48.4 ± 1.6 | 30.1 ± 4.3 | 47.4 ± 10.0 | 34.8 ± 0.8 | 51.5 ± 3.5 | 30.3 ± 4.8 | 45.2 ± 7.0 | 33.9 ± 1.9 | 50.7 ± 3.6 | 26.8 ± 2.0 | 45.5 ± 9.3 | 37.9 ± 3.2 |
| ADF (% DM) | 28.5 ± 2.1 | 25.7 ± 3.3 | 27.8 ± 1.0 | 14.0 ± 4.9 | 31.8 ± 8.3 | 21.8 ± 1.0 | 31.5 ± 3.4 | 12.4 ± 2.1 | 29.8 ± 4.5 | 20.6 ± 0.4 | 29.7 ± 2.4 | 11.4 ± 0.8 | 29.2 ± 7.7 | 26.0 ± 1.4 |
| ME (MJ/kg DM) | 10.1 ± 0.3 | 10.4 ± 0.4 | 10.1 ± 0.1 | 11.8 ± 0.6 | 9.7 ± 1.0 | 10.9 ± 0.1 | 9.7 ± 0.4 | 11.9 ± 0.2 | 9.9 ± 0.5 | 11.0 ± 0.04 | 9.9 ± 0.3 | 12.06 ± 0.1 | 10.0 ± 0.9 | 10.4 ± 0.2 |

DIM = Days in milk, NZH = New Zealand Holstein; NAH = North American Holstein; DM = dry matter; CP = crude protein; NDF = neutral detergent fiber;

ADF = acid detergent fiber; ME = metabolizable energy.

SD represents the variation between feedstuff samples.

## Plasma samples and liver biopsies

Blood samples and liver biopsies were collected at -45 ± 17, 21 ± 7, 100 ± 23 and 180 ± 23 DIM. Blood was collected by venipucture of the coccygeal vein using BD Vacutainer® tubes with heparin (Becton Dickinson) and later centrifuged at 2,000 g for 15 min at 4°C within 1 h after collection and plasma was stored at −20°C until analyses were performed. Biopsies were taken using a 14-gauge biopsy needle (Tru-Core-II Automatic Biopsy Instrument; Angiotech) after local intramuscular administration of 3 mL of 2% lidocaine hydrochloride as described previously [22] and either cryopreserved for mitochondrial oxygen consumption analyses [23] or immediately frozen in liquid nitrogen. All samples were stored at -80°C until analysis. Analysis of cryopreserved samples was performed up to two months after sampling. For western blots and qRT-PCR, 8 cows were randomly chosen within each FS and strain since these are costly and time-consuming techniques.

## Plasma and hepatic metabolites

Plasma aspartate aminotransferase and gamma-glutamyl transferase catalytic activity were determined with a commercial kit from Biosystems S.A., following manufacturer instructions as previously described [13] using a Varioskan™ Flash Multimode Reader (Thermo Fisher Scientific).

Quantification of free liver glucose and glycogen was performed in liver biopsies homogenized in 500 μL 2N HCl as previously described [8]. Briefly, homogenates were subjected to 100°C during an hour for glycogen digestion to glucose by acid-heat hydrolysis [24]. Free liver glucose and digested liver glycogen were determined using a kit from Biosystems S.A., following manufacturer instructions after neutralizing acid samples with an equal amount of 2M NaOH. Absorbance was measured at λ = 505 nm using a Multiskan™ FC Microplate Photometer (Thermo Fisher Scientific).

For liver triglyceride quantification, liver homogenates were performed according to Armour et al. [25]. In brief, liver biopsies were homogenized in lysis buffer (140 mM NaCl, 50 mM Tris and 1% Triton X-100, pH 8) and measured using a kit from Biosystems S.A., following manufacturer instructions at λ = 505 nm using a Multiskan™ FC Microplate Photometer (Thermo Fisher Scientific). For all metabolite assays intra and inter-assay coefficient of variation (CV) were less than 10%.

## Mitochondrial oxygen consumption rate

Mitochondrial respiration was studied measuring oxygen consumption rate in a high-resolution respirometer OROBOROS Oxygraph - 2k (Oroboros Instruments) at 37°C [13,23]. Briefly, electrodes were calibrated in modified MIR05 respiration medium (0.5 mM EGTA, 3mM $MgCl_2 \cdot 6H_2O$, 60 mM MOPS, 20 mM taurine, 10 mM $KH_2PO_4$, 20 mM HEPES, 110 mM sucrose, 1 g.L$^{-1}$ BSA, pH 7.1) with a saturated oxygen concentration of 191 μM at 100 kPa barometric pressure at 37°C. Respiratory rates (pmol $O_2$.min$^{-1}$.mL$^{-1}$) were calculated using the DatLab 4 analysis software (Oroboros Instruments). Liver biopsies were weighed (2–10 mg), added to the chamber and oxygen consumption measurements were obtained before and after the sequential addition of specific substrates of the respiratory chain, 10 mM glutamate and 5 mM malate (complex I) or 20 mM succinate (complex II) or 50 μM palmitoyl-CoA and 1 mM carnitine (fatty acid driven respiration) [26], followed by 4 mM adenosine diphosphate (ADP), 2 μM oligomycin (ATP synthase inhibitor), 2–4 μM carbonyl cyanide-p-trifluoromethoxyphenylhydrazone (FCCP, an uncoupler of oxidative phosphorylation). Maximum uncoupling was obtained by FCCP titration. Respiration was inhibited with 0.5 μM rotenone (complex I inhibitor) or 2.5 μM antimycin A (complex III inhibitor). After inhibiting mitochondrial respiration, diphenyliodonium (DPI) was used to inhibit NADPH oxidases [27], this was performed only during two time points, prepartum and early lactation (-45 and 21 DIM, respectively) to focus solely on the transition period. All respiratory parameters and indices were obtained as described in García-Roche et al. (2018). In brief, non-mitochondrial oxygen consumption rate measured after the addition of the specific inhibitors rotenone or antimycin A and subtracted from all other values before calculating the respiratory parameters. State 4 respiration was determined as the baseline measurement obtained with complex I and II substrates before the addition of ADP and state 3 was determined after the addition of ADP. The respiratory control ratio is the ratio between state 3 and state 4. Oligomycin-resistant respiration (ATP-independent) was measured after addition of oligomycin and oligomycin sensitive respiration (ATP-dependent) was the difference between state 3 and oligomycin-resistant respiration. Finally, the maximum respiratory rate was determined after titration with FCCP. Since cryopreserved biopsies can be stored during a limited time [23] mitochondrial respiration assays were performed in biopsies from -45, 21 and 180 DIM. In addition, due to the limited storage time and large amount of samples three treatments were studied for fatty acid driven respiration: FixP NAH, MaxP NZH and MaxP NAH and seven cows within each treatment were randomly chosen. For this, cows of the North American Holstein strain were prioritized since it is the predominant strain in the national herd [5]. Comparisons between strains and FS were performed (MaxP NZH *vs.* MaxP NAH and FixP NAH *vs.* MaxP NAH, respectively).

## Carnitine palmitoyltransferase activity

Specific activity of carnitine palmitoyltransferase (CPT) was measured by following the release of CoA-SH from palmitoyl-CoA at 412 nm using 5,5'-dithio-bis-(2-nitrobenzoic acid) (DTNB) and a molar extinction coefficient of 13600 M$^{-1}$.cm$^{-1}$ according to Bieber et al., [28]. To isolate mitochondrial fractions liver biopsies were homogenized in 0.25 M sucrose and 0.2 mM EDTA adjusted to pH 7.5 with Tris-HCl buffer with protease inhibitors (SigmaFast Protease Inhibitor Cocktail and 1 mM phenylmethylsulfonyl fluoride), homogenates were centrifuged for 13 min at 750 g and 4°C. Then, the supernatant was collected and centrifuged at 6700 g and 4°C for 12 minutes. Finally, the pellet was resuspended and washed in homogenization buffer repeating the 6700 g centrifugation. The remaining pellet was resuspended in 70 mM sucrose, 2 mM HEPES buffer at pH 7.4 and 1 mM EDTA with protease inhibitors: SigmaFast Protease Inhibitor Cocktail and 1 mM phenylmethylsulfonyl fluoride. Protein content

was determined with the Bradford assay using bovine serum albumin as standard [29]. For the assay, 116 mM Tris-HCl, pH 8.0, 0.09% Triton X-100, 1.1 mM NaEDTA, 0.035 palmitoyl-CoA, 0.12 mM DTNB, 1.1 mM L-carnitine and the mitochondrial suspension were used. For each sample, an assay was run without the addition of L-carnitine and palmitoyl-CoA to correct for unspecific reduction of DTNB. Assays were performed in duplicate using a final volume of 200 μL in a Multiskan$^{TM}$ FC Microplate Photometer (Thermo Fisher Scientific). Data is presented in units of per mg of tissue protein, considering a unit as the amount of enzyme that produces one μmol of CoA-SH per min.

## Western blots

For western blots, mitochondrial fractions were obtained as previously described by García-Roche et al., (2019) using a lysis buffer (250 mM sucrose, 50 mM Tris-HCl, 5 mM MgCl$_2$) with protease (SigmaFast Protease Inhibitor Cocktail and 1 mM phenylmethylsulfonyl fluoride) and deacetylase inhibitors, (1 μM trichostatin A and 5 mM nicotinamide, pH 7.4) and a mitochondrial resuspension buffer composed of 50 mM Tris-HCl, 1 mM EDTA, 0.5% Triton-X-100 with protease and deacetylase inhibitors, pH 6.8. Protein content was determined with the Bradford assay using bovine serum albumin as standard [29] and samples were kept at -80˚C until analyzed. Liver homogenates were resolved (30 μg) in 12% Tris-Glycine-SDS polyacrylamide gels (SDS/PAGE), along with protein ladder (#P7712, New England Biolabs) and proteins were transferred overnight to nitrocellulose membranes. Membranes were blocked with blocking buffer (Tris buffered saline with 0.1% Tween 20 and 5% skimmed milk) and incubated overnight at 4˚C with a primary antibody against acetylated lysine (1:1000, Cell Signaling Technology, 9441). For protein detection, membranes were washed and probed with secondary antibodies from LI-COR Biosciences: anti-rabbit (1:15,000, IRDye 680, 926–68071). Immunoreactive proteins were detected with an infrared fluorescence detection system (Odyssey, LI-COR Biosciences) and bands were quantified with ImageJ software by densitometry and protein levels were normalized by Ponceau staining.

## RNA isolation and qRT-PCR

Total RNA extraction from liver tissue and cDNA synthesis by reverse transcription was performed [22] using the Trizol reagent followed by lithium chloride precipitation and DNase treatment using Ambion™ DNA-free™ DNA Removal Kit (Thermo Fisher Scientific). Concentration of RNA was determined measuring absorbance at λ = 260 nm (NanoDrop ND-1000 Spectrophotometer; Nanodrop Technologies), and purity and integrity of RNA isolates were assessed from 260/280 and 260/230 absorbance ratios (greater than 1.9 and 1.8, respectively). Samples of RNA were stored at −80˚C. The SuperScript™ III Reverse Transcriptase kit (Invitrogen™ from Thermo Fisher Scientific) was used to perform retrotranscription along with random hexamers and 1 μg of total RNA as a template. The cDNA was stored at −20˚C until its use in the real-time PCR. Primers (Supplementary information 1) to specifically amplify cDNA of target genes: very long-chain acyl-CoA dehydrogenase (ACADVL), acetyl-CoA acetyltransferase 1 (ACAT1), ß-actin (ACTB), acyl-CoA oxidase 2 (ACOX2), apolipoprotein A4 (APOA4), apolipoprotein A5 (APOA5), apolipoprotein C2 (APOC2), CD36 molecule (CD36), CD40 molecule (CD40), carnitine palmitoyl-transferase 1 (CPT1A), liver fatty acid binding protein (FABP1), fibroblast growth factor 21 (FGF21), hydroxymethylglutaryl-CoA synthase 2 (HMGCS2), hypoxanthine phosphoribosyl transferase (HPRT1); nuclear receptor subfamily 1 group H member 3 (LXRA), nuclear factor kappa B subunit 1 (NFKB1), nuclear factor kappa B inhibitor alpha (NFKB1A), peroxisome proliferator-activated receptor alpha (PPARA), peroxisome proliferator-activated receptor gamma coactivator 1-alpha (PPARGC1A), retinoic acid

receptor alpha *RARA)*, retinoic X receptor alpha *(RXRA)*, retinoic X receptor beta *(RXRB)*, retinoic X receptor gamma *(RXRG)*, sterol regulatory element binding transcription factor 1 *(SREBP-1)*, tumor necrosis factor alpha *(TNFA)* and tumor necrosis factor receptor superfamily member 1A *(TNFRSF1A)* were obtained from literature or specifically designed using the Primer3 website (http://www.bioinformatics.nl/cgi-bin/primer3plus/primer3plus.cgi) and bovine nucleotide sequences available from NCBI (https://www.ncbi.nlm.nih.gov/).

Real time PCR reactions were carried out in a total volume of 15 μl using Maxima SYBR Green/ROX qPCR Master Mix 2X (Thermo Fisher Scientific), using the following standard amplification conditions: 10 min at 95˚C and 40 cycles of 15 s at 95˚C, 30 s at 60˚C, and 30s at 72˚C in a 48-well StepOne™ Real-Time PCR System (Applied Biosystems™ from Thermo Fischer Scientific). Melting curves were run on all samples to detect primer dimers, contamination, or presence of other amplicons. Each plate was designed including a pool of total RNA from bovine liver samples analyzed in triplicate to be used as the basis for the comparative expression results (exogenous control) and duplicate wells of non-template control (water). Gene expression was determined by relative quantification with respect to the exogenous control [30] and normalized to the geometric mean expression of the endogenous control genes (*ACTB* and *HPRT*). Expression stability of two selected housekeeping genes was evaluated using MS-Excel add-in Normfinder (MDL, Aarhus, Denmark), values obtained with Normfinder were for 0.004 for *ACTB* and 0.003 for *HPRT*. Amplification efficiencies or target and endogenous control genes were estimated by linear regression of a cDNA dilution curve (S1 Table). Intra and inter-assay CV values were less than 2.3 and 3.1%, (n = 5 dilutions, from 100 to 6.25 ng/well), respectively.

## Statistical analysis

Data were analyzed using the SAS software (SAS Academic Edition; SAS Institute Inc., Cary, NC, USA). Univariate and linear regression analyses were performed for all variables to identify outliers and inconsistencies and to verify normality of residuals. When data did not have normal distribution, logarithmic transformations were performed to approximate more closely to normality and homogeneity requirements. Data points with studentized residual was > 3 and < -3 were considered outliers and excluded from the analysis, no more than three values had to be excluded per variable. Least square means and pooled standard error values of all variables are presented as non-transformed data to aid in the comparison among variables.

Since the experiment consisted of four farmlets handled independently, MaxP-NZH, FixP-NZH, MaxP-NAH and FixP-NAH, with daily rotational strips and individual feeding in the milking parlor, the cow was considered the experimental unit to which the treatment was applied and repeated measures were taken of each cow (sampling timepoints were considered the observational unit) [31].

Data were analyzed as repeated measures, using the MIXED procedure, the model included the fixed effects of Holstein strain (NZH *vs.* NAH), DIM, FS (FixP *vs.* MaxP) and their interactions and block as random effect; with cows integrated as subject. Spatial power was used as covariance structure and the Kenward-Rogers procedure was used to adjust the denominator degree of freedom. Least square means tests were conducted to analyze differences between groups. For gene expression data, contrasts were performed with Tukey tests, in this case compound symmetry was used as the covariance structure. Means were considered to differ when $P < 0.05$ and a trend was declared when $0.05 < P \leq 0.10$. The CORR procedure was used to perform Pearson correlations and Spearman rankings.

The interaction between DIM, FS and strain was not significant for most variables, except for milk protein (%). For this reason, *P*-values of all fixed effects and their interactions are

presented in tables and the interaction DIM x strain and DIM x FS are presented in separate graphs in parallel panels for better interpretation. The interaction strain x FS is presented in graphs when significant. Comparisons between MaxP NZH *vs*. MaxP NAH and FixP NAH *vs*. MaxP NAH are shown in graphs when significant for variables related to gene expression.

## Results

### Milk yield and milk composition

The interaction between DIM and strain within FS was only significant for protein content in milk (Table 3, P < 0.05) as it was higher during 21 and 180 DIM for cows of the NZH strain in both FS when compared to cows of the NAH strain in the MaxP FS (3.99 and 4.01 *vs*. 3.44 ± 0.17% at 21 DIM, and 3.73 and 3.74 *vs*. 3.23 ± 0.17% at 180 DIM for NZH in MaxP and FixP and NAH in MaxP, respectively, P < 0.05).

Most variables related to milk yield and composition were affected by the interaction between DIM and strain and DIM and FS (Fig 1), except for milk yield where the interaction between DIM and strain was not observed. However, there was a main effect of strain, whereby NAH cows had higher milk yield than NZH cows (30.3 *vs*. 25.5 ± 1.3 kg.d$^{-1}$, P < 0.001, Fig 1A and Table 3). The interaction between DIM and FS (P < 0.01, Fig 1B) elicited that milk yield was higher for the MaxP than the FixP feeding strategy at 21 and 180 DIM (31.7 *vs*. 28.1 and 25.6 *vs*. 22.1 ± 1.3 kg.d$^{-1}$ for 21 and 180 DIM, respectively, P < 0.05, Fig 1B).The tendency for an interaction between DIM and strain showed that fat content was higher in NZH than NAH cows at 180 DIM (4.77 *vs*. 4.17 ± 0.15%, P < 0.01, Fig 1C). In addition, fat content in milk peaked at 21 DIM and decreased from 100 to 180 DIM for MaxP, while it increased for FixP (5.16, 4.85 and 4.34 ± 0.16% for MaxP at 21, 100 and 180 DIM and 5.11, 3.91 4.70 ± 0.16% for FixP at 21, 100 and 180 DIM, P < 0.05, Fig 1D). Protein content in milk was greater in NZH than NAH cows during 21 and 180 DIM (4.00 *vs*. 3.58 and 3.73 *vs*. 3.34 ± 0.12%, P < 0.01, Fig 1E) and was higher at 100 DIM for cows of the MaxP than FixP strategy (3.58 *vs*. 3.27± 0.09%, for, P < 0.05, Fig 1F). The SCM yield peaked at 21 DIM and was greater for the MaxP than FixP strategy (37.4 vs. 33.3 ± 1.1 kg.d$^{-1}$, P < 0.001, Fig 1H). The differences between milk yield and composition among strains translated into greater SCM yield for the NAH strain than the

**Table 3. Milk yield and composition of grazing Holstein cows of North American and New Zealand strains in two feeding strategies with varying proportions of grazed pasture during lactation.**

| | Strains[1] | | | | SEM | P-value | | | | | | |
| | NZH | | NAH | | | | | | | | | |
| | FixP | MaxP | FixP | MaxP | | DIM | Strain | FS | Strain x FS | DIM x Strain | DIM x FS | DIM x FS x Strain |
|---|---|---|---|---|---|---|---|---|---|---|---|---|
| Milk yield (kg.d$^{-1}$) | 24.1 | 26.8 | 29.1 | 31.5 | 1.3 | <0.0001 | <0.001 | <0.05 | 0.90 | 0.72 | <0.01 | 0.35 |
| Fat (%) | 4.71 | 4.82 | 4.4 | 4.68 | 0.15 | <0.0001 | 0.17 | 0.24 | 0.67 | 0.06 | <0.0001 | 0.73 |
| Protein (%) | 3.71 | 3.71 | 3.44 | 3.48 | 0.09 | <0.0001 | <0.01 | 0.83 | 0.84 | <0.01 | <0.01 | <0.05 |
| Solids corrected milk (kg.d$^{-1}$) | 23.9 | 25.8 | 24.9 | 26.6 | 0.7 | <0.0001 | 0.47 | <0.01 | 0.95 | <0.001 | <0.001 | 0.30 |

Data are shown as least square means ± standard error. DIM: days in milk; FS: feeding strategy
[1]NZH = New Zealand Holstein; NAH = North American Holstein. N = 10–12.

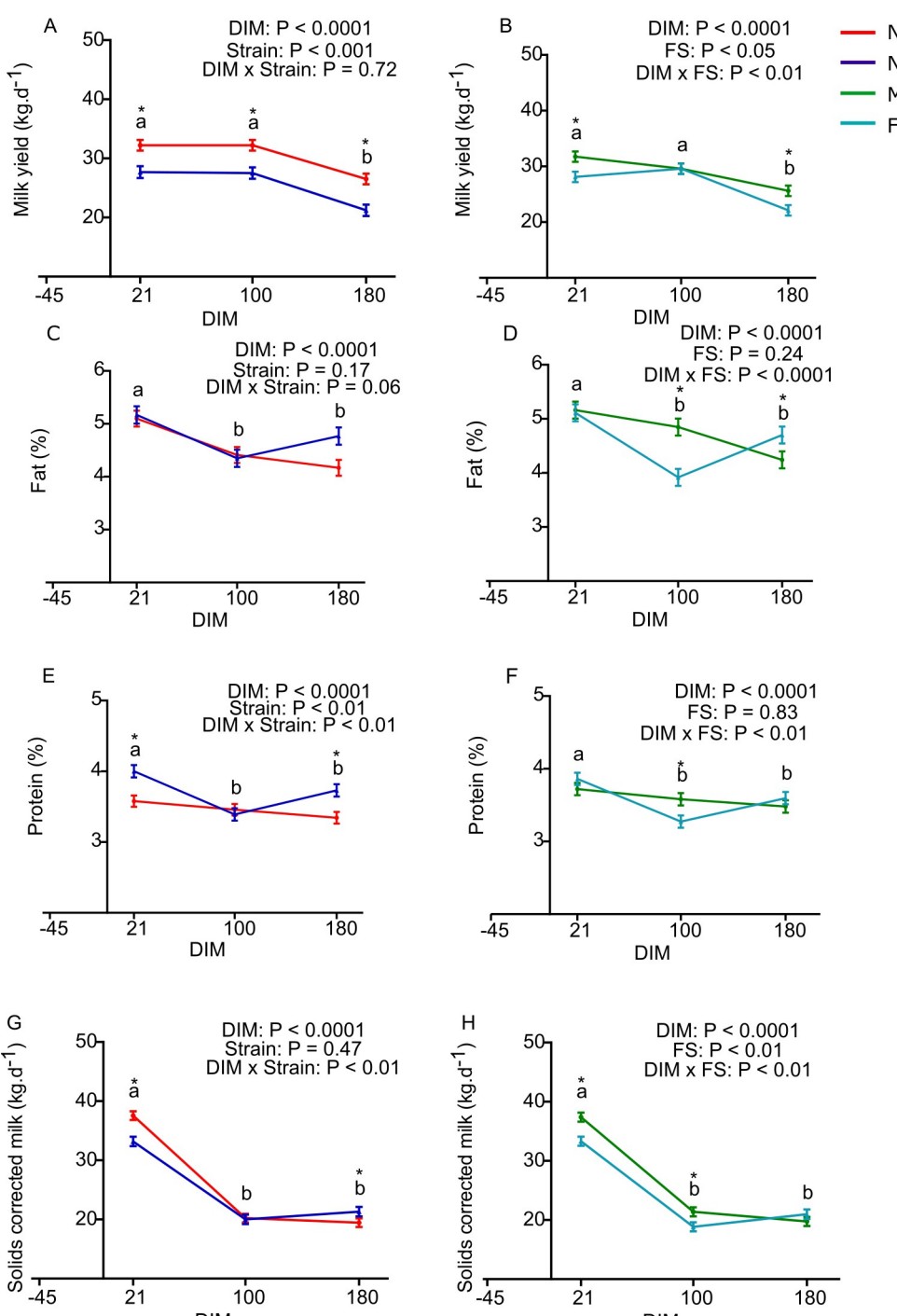

**Fig 1. Milk yield and composition of grazing Holstein cows of North American and New Zealand strains in two feeding strategies with varying proportions of grazed pasture at -45, 21, 100 and 180 DIM.** (A) and (B) Milk yield, (C) and (D) fat content in milk, (E) and (F), protein content in milk and (G) and (H) solids corrected milk. NAH = North American Holstein (red); NZH = New Zealand Holstein (blue); FixP = fixed proportion of pasture (light blue); MaxP = maximum proportion of pasture (green); DIM = days in milk; FS: feeding strategy. (N = 20–22). Different letters (a and b for P < 0.05, x and y for P ≤ 0.10) and asterisks (* for P < 0.05) depict differences between means; letters for DIM effect and asterisks for effect of the effect of strain or FS.

NZH at 21 DIM (37.5 *vs*. 33.2 and 20.8 *vs*. 18.8 ± 1.1 kg.d$^{-1}$, P < 0.05, Fig 1G). The interaction between strain and FS was not significant for any of the variables analyzed (Table 3).

## Hepatic metabolites

The interaction between DIM and strain within FS was not significant for any of the hepatic metabolites analyzed, neither were the two-way interactions between DIM and strain and DIM and FS (Table 4 and Fig 2)

Free liver glucose (P < 0.001, Fig 2A and 2B) was only affected by DIM as it peaked during pre-partum. Liver glycogen was affected by the interaction between strain and FS as it was the highest for NZH in the MaxP strategy (P< 0.05, Table 4). It was also affected by DIM, plummeting during early lactation, and increasing toward early mid-lactation (P < 0.0001, Fig 2C and 2D). Liver triglyceride levels were affected by the fixed effects DIM and FS as it increased three-fold from pre-partum to early lactation and decreased four-fold from early to mid-lactation (P < 0.0001, Fig 2E and 2F). Also, liver triglyceride levels were 50% higher for the MaxP *vs*. FixP strategy (P < 0.05, Fig 2F and Table 4). In relation to liver health markers, mean plasma levels of aspartate aminotransferase were less than 39, 69 and 71 ± 5 IU.L$^{-1}$ and mean plasma gamma-glutamyl transferase levels were less than 26, 27 and 27 ± 2 IU.L$^{-1}$ at -45, 21 and 180 DIM.

## Mitochondrial function and CPT activity

Oxygen consumption rates were measured in liver biopsies after the addition of specific substrates for mitochondrial chain complexes I and II and respiratory parameters were calculated (Figs 3A–3D and 4A–4D, respectively). Mitochondrial respiration parameters were not affected by the interaction between DIM and strain within FS or the two-way interaction between DIM and strain (S2 and S3 Tables). Complex-I dependent non-mitochondrial oxygen consumption was affected by the interaction between DIM and strain as it tended to be lower for NAH than NZH at -45 DIM and tended to be higher for NAH than NZH at 21 DIM (4.7 *vs*. 5.5 ± 0.3 and 6.8 *vs*. 5.9 ± 0.3 pmol O$_2$.min$^{-1}$.mg$^{-1}$, P = 0.06 and P = 0.08, for -45 and 21 DIM respectively; S2 Table). Oligomycin-sensitive respiration–respiration linked to ATP synthesis–driven by glutamate and malate was lower at 21 DIM for the MaxP strategy (P < 0.05,

**Table 4. Hepatic energy reserves of grazing Holstein cows of North American and New Zealand strains in two feeding strategies with varying proportions of grazed pasture during lactation.**

|  | Strains[1] | | | | SEM | *P*-value | | | | | | |
|---|---|---|---|---|---|---|---|---|---|---|---|---|
|  | NZH | | NAH | | |  |  |  |  |  |  |  |
|  | FixP | MaxP | FixP | MaxP | | DIM | Strain | FS | Strain x FS | DIM x Strain | DIM x FS | DIM x FS x Strain |
| Free liver glucose (mmol/g) | 0.025 | 0.023 | 0.020 | 0.024 | 0.003 | < 0.001 | 0.52 | 0.69 | 0.34 | 0.68 | 0.74 | 0.28 |
| Liver glycogen (m/m %) | 2.45$^{ab}$ | 2.88$^a$ | 2.45$^{ab}$ | 2.11$^b$ | 0.24 | < 0.0001 | 0.2 | 0.97 | < 0.05 | 0.18 | 0.92 | 0.21 |
| Liver triglyceride (m/m %) | 3.55 | 5.10 | 2.79 | 5.25 | 0.70 | < 0.0001 | 0.19 | < 0.01 | 0.73 | 0.82 | 0.25 | 0.24 |

Data are shown as least square means ± standard error. DIM: days in milk; FS: feeding strategy

[1]NZH = New Zealand Holstein; NAH = North American Holstein. N = 38–44.

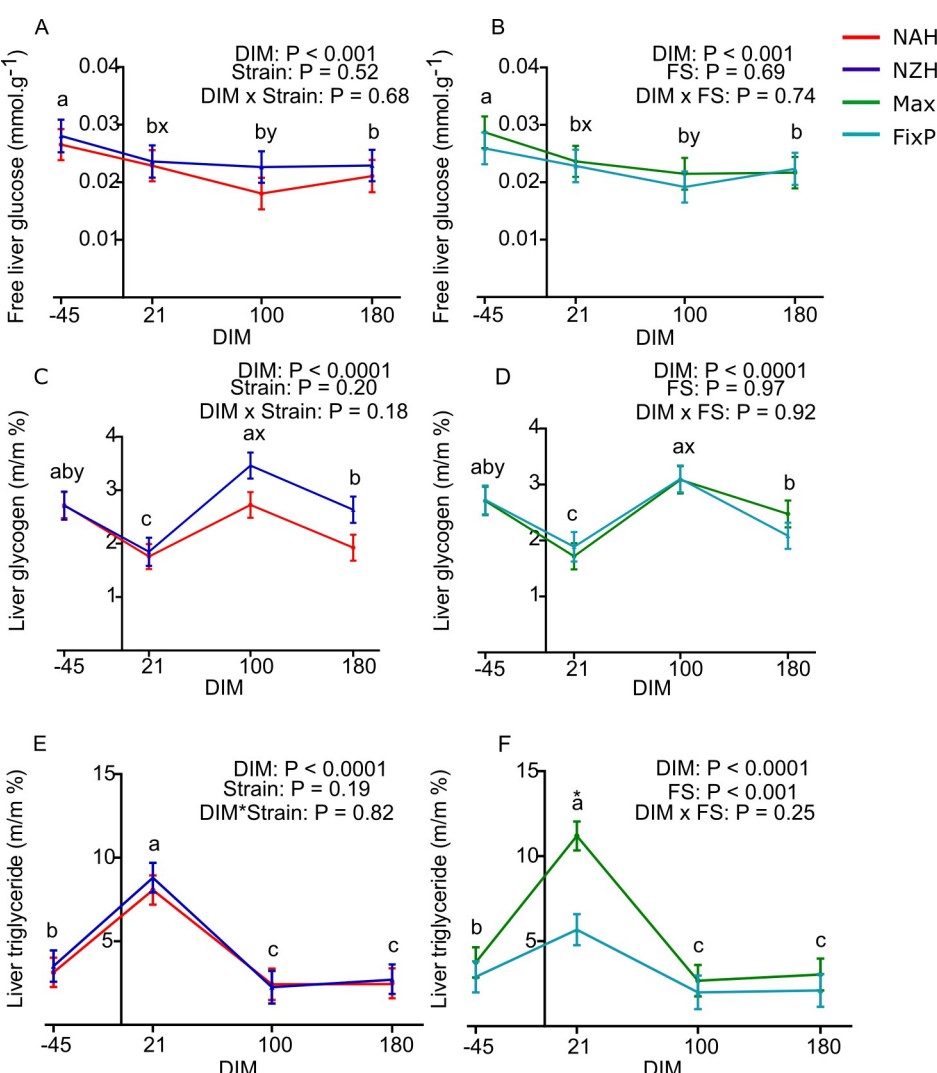

**Fig 2. Hepatic energy reserves of grazing Holstein cows of North American and New Zealand strains in two feeding strategies with varying proportions of grazed pasture at -45, 21, 100 and 180 DIM.** (A) and (B) Free liver glucose, (C) and (D), liver glycogen, (E) and (F), liver triglyceride. NAH = North American Holstein (red); NZH = New Zealand Holstein (blue); FixP = fixed proportion of pasture (light blue); MaxP = maximum proportion of pasture (green); DIM = days in milk; FS: feeding strategy. (N = 19–22). Different letters (a and b for P < 0.05, x and y for P ≤ 0.10) and asterisks (* for P < 0.05) depict differences between means; letters for DIM effect and asterisks for effect of the effect of strain or FS.

Fig 3B). The main effect of FS tended to be significant (P = 0.08) when succinate was used as substrate as it was lower in the MaxP strategy (Fig 3D).

The interaction of strain by FS was observed for the maximum respiratory capacity and respiratory control ratio (P < 0.05, Fig 4A and 4B) in the case of complex-I respiratory parameters and the highest values were observed in cows of the NZH MaxP treatment. On the other hand, the interaction between strain and FS was observed in all complex-II respiratory parameters (state 3 respiration, state 4 respiration, maximum respiratory capacity, oligomycin-resistant respiration, oligomycin-sensitive respiration and respiratory control rate, P < 0.05, Fig 4C and 4D) except non-mitochondrial respiration and the lowest values were observed in cows of

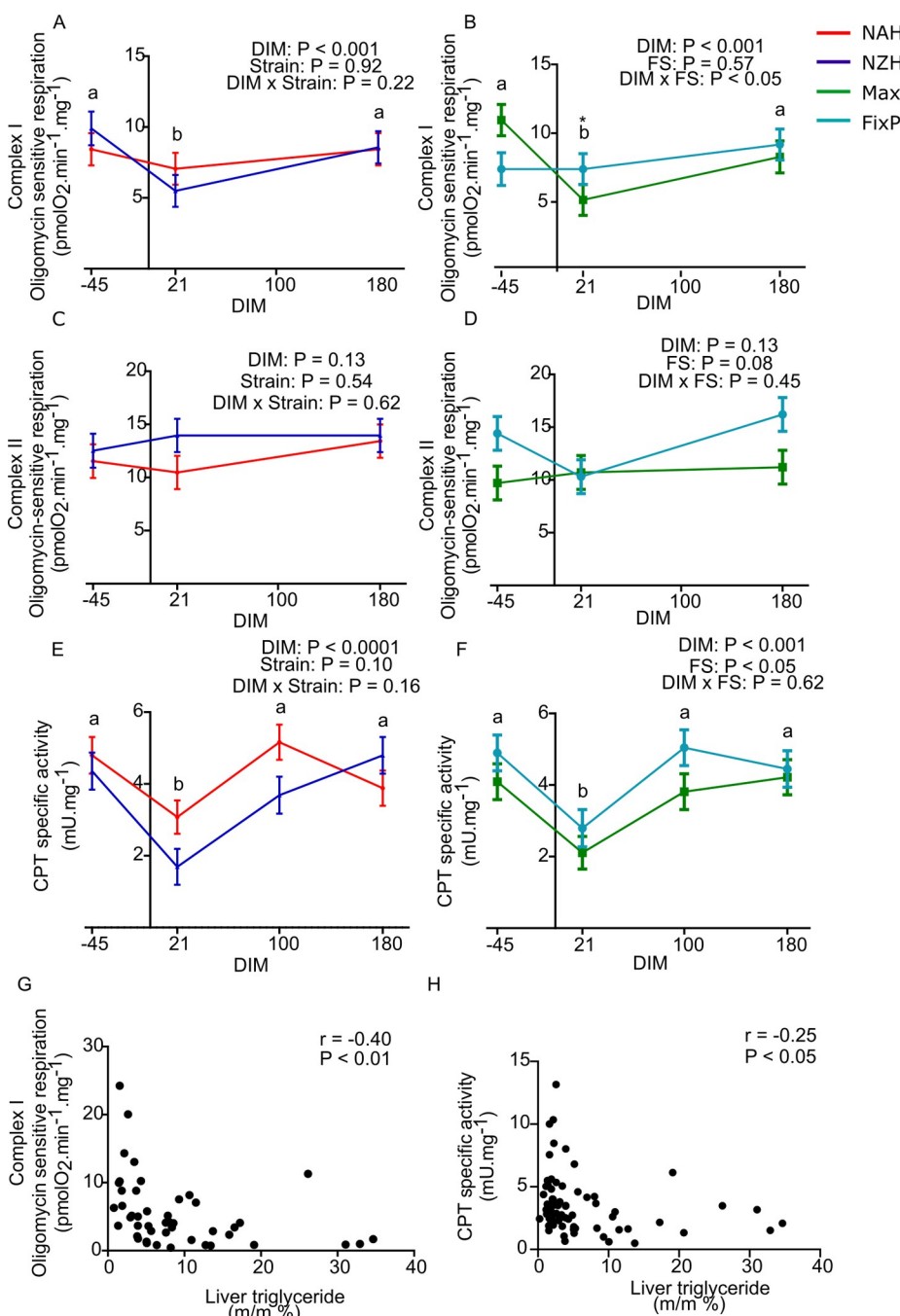

**Fig 3. Mitochondrial function and CPT activity in liver of grazing Holstein cows of North American and New Zealand strains in two feeding strategies with varying proportions of grazed pasture during lactation.** (A) and (B) Complex-I oligomycin sensitive respiration, (C) and (D) complex-II oligomycin sensitive respiration during lactation, (E) and (F) specific activity of CPT, (G) correlation between complex-I oligomycin-sensitive respiration and liver triglyceride and (H) correlation between CPT activity and liver triglyceride.NAH = North American Holstein (red); NZH = New Zealand Holstein (blue); FixP = fixed proportion of pasture (light blue); MaxP = maximum proportion of pasture (green); DIM = days in milk; FS: feeding strategy. (N = 18–22). Different letters (a and b for P < 0.05) and asterisks (* for P < 0.05) depict differences between means; letters for DIM effect and asterisks for effect of the effect of strain or FS.

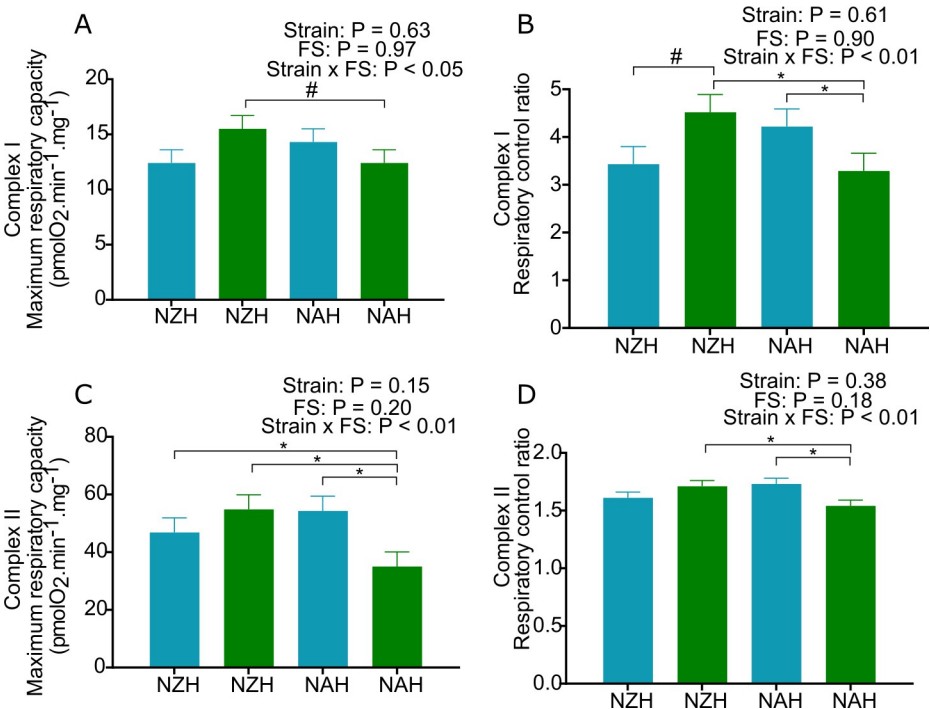

**Fig 4. Effect of feeding strategy and Holstein strain on hepatic mitochondrial function.** (A) complex-I maximum respiratory capacity, (B) complex-I respiratory control ratio, (C) complex-II maximum respiratory capacity, (D) complex-II respiratory control ratio. NAH = North American Holstein; NZH = New Zealand Holstein. Feeding strategies (FS): FixP = fixed proportion of pasture (light blue) and MaxP = maximum proportion of pasture (green); DIM = days in milk; FS: feeding strategy. (N = 40–44). * for P < 0.05 and # for 0.05 < P < 0.10.

the NAH strain in the MaxP strategy. All complex I and II respiratory parameters are reported in S2 and S3 Tables.

Specific activity of CPT was not affected by the interaction between DIM and strain within FS or any of the two-way interactions (S4 Table). It tended to be affected by strain (4.2 *vs.* 3.6 ± 0.3 mU.mg$^{-1}$ for NAH and NZH, respectively, P = 0.10 Fig 3E). Specific CPT activity was affected by the fixed effects DIM and FS since it was lower at 21 DIM (2.4 *vs.* 4.5, 4.4 and 4.3 ± 0.3 mU.mg$^{-1}$ for 21, -45, 100 and 180 DIM, respectively, P < 0.001, Fig 3E) and was lower for the MaxP than the FixP strategy (4.3 *vs.* 3.6 ± 0.3 mU.mg$^{-1}$ for FixP and MaxP respectively, P < 0.05, Fig 3F).

Correlations were performed between complex-I oligomycin-sensitive respiration and hepatic energy reserves at 21 and 180 DIM and a negative association was found between oligomycin-sensitive respiration and liver triglyceride (r = -0.40, P < 0.01 for Pearson correlation, Fig 3G), while no association was found between oligomycin-sensitive respiration and liver glycogen (r = 0.032, P = 0.86 for Pearson correlation). Also, the correlation between CPT activity and liver triglyceride was negative (r = -0.25, P < 0.05 for Pearson correlation, Fig 3H).

To assess oxygen consumption from NADPH oxidases, non-mitochondrial oxygen consumption measured after inhibition of complexes I and III with the addition of rotenone and antimycin A, respectively was further inhibited with the addition of DPI at -45 and 21 DIM (Table 5). The interaction between DIM and strain within FS and the two-way interactions between DIM and strain, DIM and FS and strain by FS were not significant. However, the fraction of non-mitochondrial oxygen consumption inhibited by DPI was higher for NAH *vs.* NZH cows (0.65 *vs.* 0.55 ± 0.03, P < 0.01).

**Table 5. Inhibition of non-mitochondrial oxygen consumption rate by diphenyliodonium in liver biopsies of North American (NAH) and New Zealand (NZH) grazing Holstein cows with varying proportions of grazed pasture during prepartum and early lactation.**

| DIM | Treatments | | | | SEM | P-value | | | | | | |
| | FixP | | MaxP | | | DIM | Strain | FS | DIM x Strain | DIM x FS | Strain x FS | DIM x FS x Strain |
| | NZH | NAH | NZH | NAH | | | | | | | | |
|-----|------|------|------|------|-----|-----|--------|------|---------|---------|---------|---------|
| -45 | 0.54 | 0.67 | 0.54 | 0.60 | 0.03 | 0.62 | < 0.01 | 0.55 | 0.38 | 0.61 | 0.15 | 0.97 |
| 21 | 0.54 | 0.65 | 0.58 | 0.62 | | | | | | | | |

Data are shown as least square means ± standard error. N = 10–12. DIM: days in milk., FS: feeding strategy.

FixP = fixed proportion of pasture, MaxP = maximum proportion of pasture.

## Mitochondrial protein acetylation

The interaction between DIM and strain within FS and the two-way interactions between DIM and strain, DIM and FS were not significant for levels of acetylated lysine in mitochondrial fractions of liver biopsies. However mitochondrial protein acetylation was affected by the interaction between strain and FS and the group with the highest levels of acetylated lysine were NAH cows in the MaxP strategy (P < 0.05, Fig 5A and 5B). We found a correlation between complex-I oligomycin-sensitive respiration and mitochondrial acetylation only when NAH cows were included in the analysis, it was r = - 0.35 and P = 0.06. All western blot membranes can be found in S1 File.

## Hepatic fatty-acid driven respiration

Fatty-acid driven respiration was also studied at -45, 21 and 180 DIM for three treatments NZH MaxP, NAH MaxP and NAH FixP. Respiratory parameters such as state 3, state 4, oligomycin-resistant sensitive respiration and the maximum respiratory rate increased as lactation progressed (Tables 6 and 7, P < 0.05). An interaction between DIM and strain and DIM and FS was found for the respiratory control ratio as it decreased for NAH MaxP cows from -45 to 180 DIM (Table 6, P < 0.05) and as it was the highest for NAH FixP cows at 21 DIM (Table 7, P < 0.01). Also, an interaction between DIM and FS was found for the maximum respiratory rate and it showed that it was 2.6-fold higher for NAH MaxP than NAH FixP cows at 180 DIM (Table 7, P < 0.01).

## Hepatic gene expression of fatty acid metabolism and transcription factors

To further explore the differences observed in hepatic fatty acid driven respiration at 180 DIM, hepatic expression of genes related to fatty acid metabolism and relevant transcription factors was assessed at 180 DIM in cows of NZH and NAH strains in the MaxP strategy and in cows of the NAH strain in the FixP and MaxP strategies. Hepatic gene expression of *ACADVL*, *RARA* and *RXRB* was at least 80% greater (P < 0.05) for NZH than NAH cows (Fig 6A, 6E and 6F and Table 8) and expression of *ACAT1*, *CD40* and *PPARA* tended (P < 0.10) to be greater for NZH than NAH cows (Fig 6B–6D and Table 8). Hepatic expression of *CD36* was almost 3-fold greater (P < 0.05) for FixP than MaxP cows (Fig 6G and Table 8). Abundance of mRNA of *ACOX2*, *APOA4*, *APOA5*, *APOC2*, *CPT1A*, *FABP1*, *FGF21*, *HMGCS2*, *LXRA*, *NFKB1*, *NFKB1A*, *PPARGC1A*, *RXRA*, *RXRG*, *SREBP-1*, *TNFA*, and *TNFRSF1A* did not differ among strains or strategies.

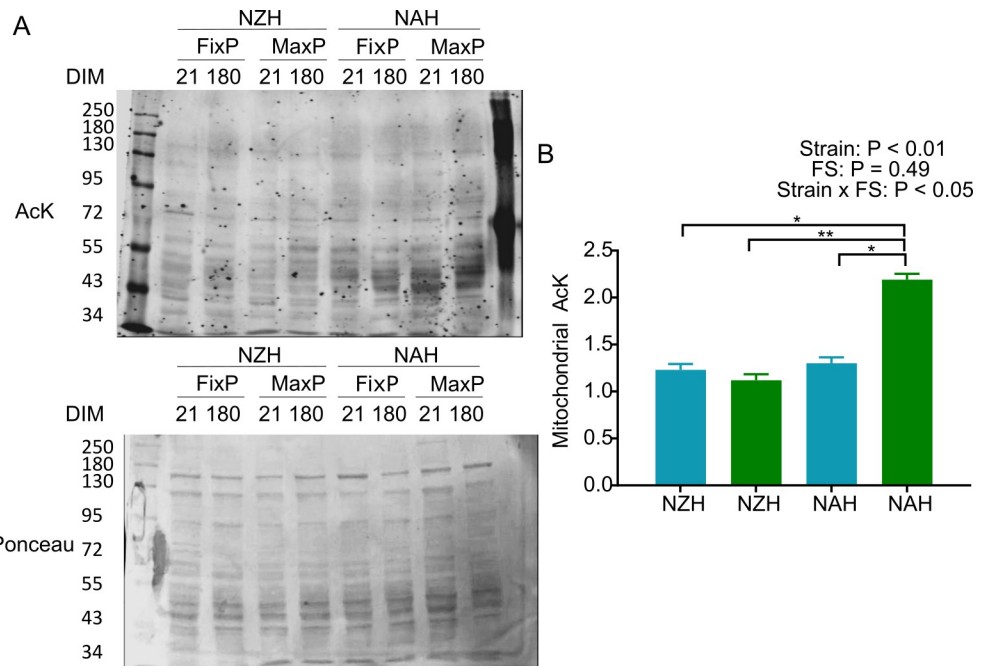

**Fig 5. Effect of feeding strategy and Holstein strain on mitochondrial protein acetylation.** A) Representative western blots of mitochondrial acetylated lysine (AcK) and in mitochondrial fractions North American (NAH) and New Zealand (NZH) grazing Holstein cows with a fixed proportion of pasture (FixP, light blue) or maximum proportion of pasture (MaxP, green) at 21 and 180 DIM (N = 8) the last lane is acetylated BSA, bottom panel shows ponceau S staining for protein normalization. (B) Western blots were quantified by densitometry, normalized with the loading control and expressed in relation to the average value of the NZH-FixP treatment at 21 DPP. * for P ≤ 0.05 and ** for P < 0.001, * depict differences between means. FS: feeding strategy.

## Discussion

The objective of this study was to characterize hepatic metabolic adaptations in Holstein cows of two strains (NAH *vs.* NZH) under two pasture-based feeding strategies that differed in the proportion of grazed pasture.

### Effect of strain on lactation

Our results showed cows of the NAH and NZH strains, independent of feeding strategy, adapted their hepatic metabolism similarly across lactation. Related to productive performance, we found NAH cows decreased the content of fat and protein in milk toward late mid-lactation and NZH cows maintained similar solid yield during mid-lactation. In addition, milk yield was higher in NAH than NZH cows during the whole trial, which is in agreement with previous studies [5,32,33].

With respect to hepatic mitochondrial respiratory parameters, we found that although NZH cows tended to have higher complex-I non-mitochondrial respiration during the prepartum, NAH cows tended to increase this parameter toward early lactation. Non-mitochondrial respiration is related to oxygen consuming processes independent of the respiratory chain, (such as NADPH oxidases). This enzyme catalyzes the one-electron reduction of molecular oxygen to superoxide anion, that can lead to the formation of ROS [34]. In this sense, we observed that a greater proportion of non-mitochondrial oxygen consumption was inhibited by diphenyliodonium, suggestive of an increase in the activity of NADPH oxidases in cows of the NAH strain, with respect to the NZH strain. This enzyme participates in innate immunity

**Table 6. Hepatic fatty acid driven respiration in liver biopsies of North American (NAH) and New Zealand (NZH) grazing Holstein cows in a pasture-based system from prepartum to late mid-lactation.**

| Respiratory parameters[1] | DIM | Strains | | SEM | P-value | | |
|---|---|---|---|---|---|---|---|
| | | NZH | NAH | | DIM | Strain | DIM x Strain |
| State 3 respiration | -45 | 5.7 | 6.2 | 1.1 | < 0.001 | 0.65 | 0.99 |
| | 21 | 6.3 | 6.3 | | | | |
| | 180 | 10.1 | 10.5 | | | | |
| State 4 respiration | -45 | 3.5 | 3.4 | 1 | < 0.01 | 0.76 | 0.57 |
| | 21 | 4 | 4.3 | | | | |
| | 180 | 8.1 | 7.4 | | | | |
| Maximum respiratory capacity | -45 | 6.1 | 7.3 | 1.4 | < 0.01 | 0.08 | 0.85 |
| | 21 | 6.4 | 8.8 | | | | |
| | 180 | 10.8 | 13.6 | | | | |
| Oligomycin-resistant respiration | -45 | 3.8 | 4.4 | 0.9 | < 0.05 | 0.49 | 0.98 |
| | 21 | 4.2 | 4.9 | | | | |
| | 180 | 6.3 | 6.6 | | | | |
| Oligomycin-sensitive respiration | -45 | 1.9 | 1.8 | 0.5 | < 0.001 | 0.84 | 0.95 |
| | 21 | 1.7 | 1.4 | | | | |
| | 180 | 3.8 | 3.9 | | | | |
| Non-mitocondrial respiration | -45 | 7.9[a] | 5.9[b] | 0.7 | 0.47 | 0.26 | < 0.05 |
| | 21 | 5.7[b] | 6.9[ab] | | | | |
| | 180 | 7.8[a] | 6.2[ab] | | | | |
| Respiratory control ratio | -45 | 1.8[ab] | 1.9[a] | 0.1 | < 0.01 | 0.91 | < 0.05 |
| | 21 | 1.8[ab] | 1.4[b] | | | | |
| | 180 | 1.2[b] | 1.4[b] | | | | |

Data are shown as least square means ± standard error. a,b denote differences between means (P < 0.05). N = 7. DIM: days in milk.

[1]Oxygen consumption rate measurements of liver biopsies were obtained after addition of 50 μM palmitoyl-CoA and 1 mM carnitine, 4 μM ADP, 2 μM oligomycin, up to 4 μM carbonyl cyanide-p- trifluoromethoxyphenylhydrazone and 0.5 μM rotenone, and 2.5 μM antimycin. Oxygen consumption rates are expressed as pmol of $O_2$ per min per mg of wet weight.

and increases in systemic inflammation [27]. Hepatic non-mitochondrial oxygen consumption has been observed in early lactation in dairy cows and associated with an increase in oxidative stress markers [13]. An increase in oxidant species derived from NADPH oxidases by immune cells may contribute to the development of pathologies such as fatty liver [35].

In dairy cows, fatty liver is a metabolic disorder which occurs in the transition period and is characterized by excessive accumulation of hepatic triglycerides, tissue dysfunction and cell death; oxidative stress and a pro-apoptotic and pro-inflammatory environment have been proposed as plausible causal factors [36].

Since our study suggested that NAH cows could present a favorable environment for the progression of fatty liver, we studied the molecular mechanisms that explained hepatic lipid accumulation in both strains in MaxP as it was the FS with higher levels of liver triglycerides. Thus, we analyzed gene expression of key enzymes involved in fatty acid oxidation during late mid-lactation to focus on the effect of strain and not early lactation NEB. We demonstrated that NZH cows had greater abundance of *ACADVL* mRNA and transcription factors associated to mitochondrial and peroxisomal fatty acid oxidation *RARA*, *RXRB* and tended to have greater mRNA abundance of *ACAT1* and *PPARA*. The transcription factor PPARα is a nuclear

**Table 7. Hepatic fatty acid driven respiration in liver biopsies of North American (NAH) grazing Holstein cows with varying proportions of grazed pasture from prepartum to late mid-lactation.**

| Respiratory parameters[1] | DIM | FS | | SEM | P-value | | |
|---|---|---|---|---|---|---|---|
| | | FixP | MaxP | | DIM | FS | DIM x FS |
| State 3 respiration | -45 | 5.8 | 6.2 | 1 | < 0.05 | 0.48 | 0.16 |
| | 21 | 7.7 | 6.3 | | | | |
| | 180 | 7.4 | 10.5 | | | | |
| State 4 respiration | -45 | 3.1 | 3.4 | 0.6 | < 0.001 | 0.07 | 0.57 |
| | 21 | 3.5 | 4.4 | | | | |
| | 180 | 5.6 | 7.4 | | | | |
| Maximum respiratory capacity | -45 | 8.1[b] | 7.2[b] | 1.7 | 0.6 | 0.1 | < 0.05 |
| | 21 | 8.6[b] | 8.7[b] | | | | |
| | 180 | 5.3[b] | 13.7[a] | | | | |
| Oligomycin-resistant respiration | -45 | 3.9 | 4.4 | 0.8 | 0.47 | 0.06 | 0.47 |
| | 21 | 4.4 | 5.1 | | | | |
| | 180 | 4.02 | 6.4 | | | | |
| Oligomycin-sensitive respiration | -45 | 1.9 | 1.4 | 0.5 | < 0.001 | 0.81 | 0.68 |
| | 21 | 2.0 | 1.9 | | | | |
| | 180 | 3.6 | 3.9 | | | | |
| Non-mitocondrial respiration | -45 | 5.1 | 6.0 | 0.8 | 0.25 | 0.998 | 0.595 |
| | 21 | 6.8 | 6.7 | | | | |
| | 180 | 7.1 | 6.4 | | | | |
| Respiratory control ratio | -45 | 1.8[b] | 1.9[ab] | 0.2 | < 0.01 | 0.2 | < 0.01 |
| | 21 | 2.2[a] | 1.5[c] | | | | |
| | 180 | 1.3[c] | 1.5[c] | | | | |

Data are shown as least square means ± standard error. a,b denote differences between means (P < 0.05), x,y denote tendencies (0.05 < P < 0.1). N = 7. FS: feeding strategy; DIM: days in milk: FixP = fixed proportion of pasture; MaxP = maximum proportion of pasture.

[1]Oxygen consumption rate measurements of liver biopsies were obtained after addition of 50 µM palmitoyl-CoA and 1 mM carnitine, 4 µM ADP, 2 µM oligomycin, up to 4 µM carbonyl cyanide-p- trifluoromethoxyphenylhydrazone and 0.5 µM rotenone, and 2.5 µM antimycin. Oxygen consumption rates are expressed as pmol of $O_2$ per min per mg of wet weight.

hormone receptor activated by fatty acids, once activated it forms a heterodimer with RXR and promotes the expression of genes that mediate fatty acid oxidation [37], among its target genes is *ACADVL*, the gene that codes for very long-chain acyl-CoA dehydrogenase a rate limiting enzyme in mitochondrial fatty acid oxidation [38]. Also, a recent study suggested that inhibition of retinoid X receptor function is the most prominently enriched pathway in liver of NEB cows accounting for impaired anti-inflammatory cytokine transcription [39]. Cytokines such as TNFA may induce extensive cell death and greater *TNFA* mRNA has been found in dairy cows with moderate fatty liver [36]. However, in this work we did not find differences in *TNFA* mRNA abundance, probably due to the fact that we studied gene expression during 180 DIM when liver triglyceride levels were the lowest.

The fact that the expression of genes related to fatty acid oxidation was downregulated in NAH cows with respect to NZH cows further contributes to the development of liver metabolic disorders in NAH cows, suggesting NZH cows have the necessary machinery to cope with excessive accumulation of liver triglycerides.

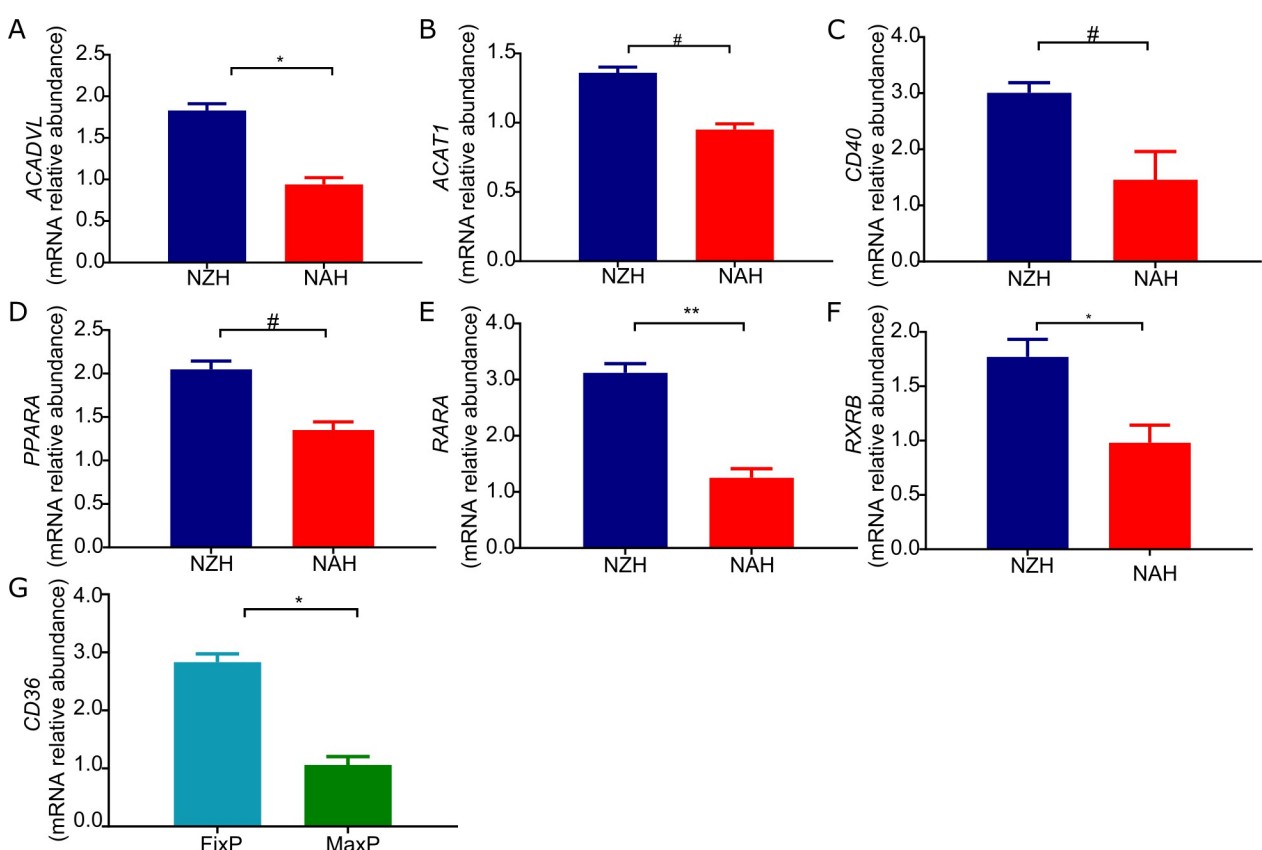

**Fig 6. Hepatic gene expression of fatty acid metabolism and transcription factors.** (A) Relative mRNA abundance of *ACADVL* in liver biopsies of North American (red, NAH) and New Zealand (blue, NZH) Holstein cows in the MaxP strategy at 180 DIM (N = 8). (B) Relative mRNA abundance of *ACAT1* in liver biopsies of North American (red, NAH) and New Zealand (blue, NZH) Holstein cows in the MaxP strategy at 180 DIM (N = 8). (C) Relative mRNA abundance of *CD40* in liver biopsies of North American (red, NAH) and New Zealand (blue, NZH) Holstein cows in the MaxP strategy at 180 DIM (N = 8). (D) Relative mRNA abundance of *PPARA* in liver biopsies of North American (red, NAH) and New Zealand (blue, NZH) Holstein cows in the MaxP strategy at 180 DIM (N = 8). (E) Relative mRNA abundance of in liver biopsies of North American (red, NAH) and New Zealand (blue, NZH) Holstein cows in the MaxP strategy at 180 DIM (N = 8). (F) Relative mRNA abundance of *RXRB* in in liver biopsies of North American (red, NAH) and New Zealand (blue, NZH) Holstein cows in the MaxP strategy at 180 DIM (N = 8). (G) Relative mRNA abundance of *CD36* in liver biopsies from cows of the NAH strain in the FixP (light blue) or MaxP (green) feeding strategy at 180 DIM (N = 8). Asterisks and numerals depict differences in means from Tukey tests * for P < 0.05, ** for P < 0.001, and # for 0.05 < P ≤ 0.10

## Effect of feeding strategy on lactation

In this study, we confirmed hepatic mitochondrial function was impaired during early lactation and its association with liver triglyceride levels in cows under the strategy with greater inclusion of grazed pasture–MaxP. In addition, we found CPT activity was decreased in this feeding strategy and found a negative correlation between CPT activity and liver triglyceride levels.

Hepatic liver triglyceride levels indicated that during early lactation cows in the MaxP strategy reached levels close to clinical fatty liver (> 10% liver triglyceride wet weight) [40]. Consistent with this, previous authors have demonstrated that plasma NEFA and beta-hydroxybutyrate levels are similar among strains [17,33], suggesting mobilization is similar among strains. Moreover, a previous report of our research group indicated that plasma NEFA were not affected by Holstein strain but by the interaction between DIM and FS as NEFA levels increased at 21 DIM for both FS but decreased more in the FixP than the MaxP strategy at 100 DIM, and coincidently plasma insulin mirrored plasma NEFA levels [41]. Indeed, early

**Table 8. Hepatic gene expression of North American (NAH) and New Zealand (NZH) grazing Holstein cows in a pasture-based system and of NAH cows in two feeding strategies with varying proportions of grazed pasture at 180 DIM.**

| Gene[1] | Treatments[2] | | | SEM | P-value | |
|---|---|---|---|---|---|---|
| | FixP | MaxP | | | NZH *vs.* NAH | FixP *vs.* MaxP |
| | NAH | NZH | NAH | | | |
| *ACADVL* | 0.89 | 1.83 | 0.94 | 0.23 | 0.04 | 0.99 |
| *ACAT1* | 1.07 | 1.36 | 0.95 | 0.12 | 0.08 | 0.79 |
| *ACOX2* | 1.80 | 2.17 | 1.15 | 0.70 | 0.48 | 0.89 |
| *APOA4* | 1.15 | 1.91 | 1.62 | 0.28 | 0.78 | 0.50 |
| *APOA5* | 1.56 | 1.71 | 1.65 | 0.73 | 0.99 | 0.98 |
| *APOC2* | 2.45 | 2.41 | 2.36 | 0.72 | 0.87 | 0.96 |
| *CD36* | 2.83 | 1.07 | 1.06 | 0.46 | 0.99 | 0.03 |
| *CD40* | 1.42 | 3.01 | 1.46 | 0.50 | 0.098 | 0.99 |
| *CPT1A* | 0.69 | 0.87 | 0.91 | 0.28 | 0.99 | 0.78 |
| *FABP1* | 2.17 | 1.27 | 1.82 | 0.70 | 0.63 | 0.83 |
| *FGF21* | 2.06 | 2.35 | 2.76 | 0.70 | 0.71 | 0.96 |
| *HMGCS2* | 1.27 | 1.68 | 0.94 | 0.28 | 0.17 | 0.68 |
| *LXRA* | 2.50 | 1.52 | 1.77 | 0.61 | 0.88 | 0.36 |
| *NFKB1* | 1.87 | 2.10 | 1.77 | 0.46 | 0.59 | 0.97 |
| *NFKBIA* | 1.52 | 2.21 | 1.57 | 0.51 | 0.17 | 0.99 |
| *PPARA* | 1.08 | 2.05 | 1.35 | 0.27 | 0.08 | 0.76 |
| *PPARGC1A* | 0.27 | 0.44 | 0.42 | 0.08 | 0.98 | 0.49 |
| *RARA* | 1.64 | 3.12 | 1.25 | 0.47 | 0.01 | 0.54 |
| *RXRA* | 2.16 | 2.23 | 1.96 | 0.70 | 0.92 | 0.96 |
| *RXRB* | 1.71 | 1.77 | 0.98 | 0.46 | 0.04 | 0.18 |
| *RXRG* | 1.28 | 2.03 | 2.28 | 0.44 | 0.93 | 0.15 |
| *SREBP-1* | 2.12 | 1.63 | 1.30 | 0.56 | 0.86 | 0.40 |
| *TNFA* | 1.53 | 1.38 | 1.31 | 0.38 | 0.99 | 0.91 |
| *TNFRSF1A* | 1.48 | 0.99 | 1.47 | 0.39 | 0.40 | 0.99 |

[1]Genes: Very long-chain acyl-CoA dehydrogenase *(ACADVL)*, acetyl-CoA acetyltransferase 1 *(ACAT1)*, ß-actin *(ACTB)*, acyl-CoA oxidase 2 *(ACOX2)*, apolipoprotein A4 *(APOA4)*, apolipoprotein A5 *(APOA5)*, apolipoprotein C2 *(APOC2)*, CD36 molecule *(CD36)*, CD40 *molecule (CD40)*, carnitine palmitoyl-transferase 1 *(CPT1A)*, liver fatty acid binding protein *(FABP1)*, fibroblast growth factor 21 *(FGF21)*, hydroxymethylglutaryl-CoA synthase 2 *(HMGCS2)*, hypoxanthine phosphoribosyl transferase *(HPRT1);* nuclear receptor subfamily 1 group H member 3 *(LXRA)*, nuclear factor kappa B subunit 1 *(NFKB1)*, nuclear factor kappa B inhibitor alpha *(NFKB1A)*, peroxisome proliferator-activated receptor alpha *(PPARA)*, peroxisome proliferator-activated receptor gamma coactivator 1-alpha *(PPARGC1A)*, retinoic acid receptor alpha *RARA)*, retinoic X receptor alpha *(RXRA)*, retinoic X receptor beta *(RXRB)*, retinoic X receptor gamma *(RXRG)*, sterol regulatory element binding transcription factor 1 *(SREBP-1)*, tumor necrosis factor alpha *(TNFA)*, tumor necrosis factor receptor superfamily member 1A *(TNFRSF1A)*.
[2]NAH–FixP = North American Holstein in the feeding strategy with fixed proportion of pasture (N = 8); NAH–MaxP = North American Holstein in the feeding strategy with maximum proportion of pasture (N = 8); NZH–MaxP = New Zealand Holstein in the feeding strategy with maximum proportion of pasture (N = 8). All data is shown as least square means ± standard error.

lactation NEB markers have been shown to be higher in pasture-based systems when compared to TMR systems [8] and this can be explained by insufficient DM intake [42]. For this reason, the use of partial mixed rations–such as the FixP feeding strategy–have been shown to improve DM intake and consequently reduce the magnitude of early lactation NEB [43].

Mitochondria play an essential role in energy production from nutrient oxidation pathways such as glycolysis and mitochondrial beta-oxidation, being the latter especially relevant in the liver when there are high levels of circulating NEFAs [38]. Herein we found hepatic mitochondrial function measured as oligomycin-sensitive respiration was impaired during early lactation and it was negatively associated with liver triglyceride. In addition, FixP cows maintained higher levels of oligomycin-sensitive respiration than MaxP cows. Differences were more significant in complex-I driven respiration, which was consistent with our previous work showing that mitochondrial function was impaired during early lactation in cows under a pasture-based system [13].

To understand the molecular mechanism underlying hepatic lipid accumulation we studied CPT activity. The CPT system is composed by three enzymes: CPT1 which catalyzes the formation of acylcarnitine from acyl-CoA and free carnitine, carnitine-acylcarnitine translocase and CPT2 which catalyzes the transport of acylcarnitines across the inner mitochondrial membrane; CPT1 is the rate limiting step in long chain fatty acid mitochondrial beta-oxidation [44]. Hence, in addition to decreased mitochondrial respiration, we also found CPT activity was decreased in cows in the MaxP *vs*. FixP strategy and a negative association with liver triglyceride levels. Contrary to our results, previous studies have suggested CPT1 activity correlates positively with liver triglyceride [44], *CPT1A* mRNA is responsive to elevated levels of hepatic lipid accumulation in calf hepatocytes [45] and *CPT1B* is upregulated in severe NEB cows [46]. However, Li et al., (2012) have shown that abundance of *CPT1* mRNA and abundance of *CPT2* mRNA and its protein levels are decreased in hepatic biopsies of ketotic *vs*. non-ketotic cows.

Decreased mitochondrial respiration and CPT activity during early lactation when plasma NEFA and liver triglyceride levels are the highest contributes to a more aggravated NEB.

To further explore the interrelation between mitochondrial respiration, CPT activity and liver triglyceride accumulation we studied fatty acid driven respiration and in general terms it mimicked complex-I and II respiratory parameters. However, the maximum respiratory capacity increased toward 180 DIM in NAH MaxP *vs*. NAH FixP cows. We studied gene expression of transcription factors and enzymes related to fatty acid oxidation between NAH MaxP *vs*. NAH FixP cows in 180 DIM and did not find any differences. However, NAH FixP cows had the lowest triglyceride to glycogen ration, high levels of glycogen reserves are related associated to lower levels of mitochondrial beta-oxidation [47]. Previous *in vitro* studies have shown that propionate–the main precursor for glucose synthesis in lactating cows–decreases palmitate oxidation in liver from dairy cows [48]. In addition, we found that abundance of *CD36* mRNA was differentially expressed in liver biopsies of NAH cows in the FixP *vs*. NAH cows in the MaxP strategy, recent evidence in mice overexpressing *CD36* has suggested that CD36 functions as a protective metabolic sensor in the liver under lipid overload and metabolic stress, promoting glycogen synthesis [49,50].

### Effect of feeding strategy and strain

In the present study, the interaction between strain and FS showed that hepatic glycogen and mitochondrial function were the lowest in NAH cows in the MaxP strategy.

The distinct adaptability to pasture-based systems in cows of the NAH and NZH strains has been widely reported, as authors have shown that cows of the NAH fail to maintain BCS and BW in pasture-based systems [5,16]. Previous reports have suggested that NAH cows have a higher maintenance requirement [14], greater uncoupling of the somatotropic axis which accounts for a greater proportion of nutrients partitioning toward milk yield at the expense of body reserves [17] and increased gluconeogenesis in detriment of TCA cycling [18]. Indeed,

mitochondria are at the crossroads of energy metabolism as they convert energy from nutri-ents to utilizable energy for cellular functions [10] and the liver in particular is a highly meta-bolic organ involved in service functions and represents an important proportion of basal energy expenditure [51]. It has been well established that the hepatocyte increases its size to meet physiological demands [52], in fact when comparing NAH *vs*. NZH cows in the MaxP feeding strategy during late mid-lactation, NZH cows showed increased hepatic mitochondrial density [18]. This evidence suggests that although NZH and NAH cows have similar levels of NEB markers, their hepatic intermediary metabolism adapts differently. In this sense, dysfunc-tional mitochondria could account not only for inefficient energy conversion but also for a propensity to lipid, protein and DNA oxidative damage [10]. In fact, recent proteomic evi-dence have confirmed that oxidative phosphorylation and mitochondrial dysfunction are rele-vant pathways in the pathogenesis of NEB [39,53]. Herein, the interaction between strain and FS also showed that hepatic mitochondrial acetylation levels were greater in NAH cows in the MaxP strategy compared to all other treatments: NAH cows in the FixP strategy, and NZH cows in the MaxP and FixP strategies. We have previously reported an association between mitochondrial function impairment and mitochondrial acetylation in early lactation, espe-cially aggravated in cows in a pasture-based system *vs*. cows fed TMR, as well as a negative association between mitochondrial deacetylase sirtuin 3 protein levels and mitochondrial acet-ylation [13]. The present work shows that levels of mitochondrial acetylation are lower in NAH cows in a pasture-based system with TMR supplementation (FixP) when compared to NAH cows in a pasture-based system with concentrate supplementation in the milking parlor (MaxP). Even though we did not confirm an effect of stage of lactation in this study, we did confirm that plane of nutrition may affect mitochondrial function and mitochondrial protein acetylation; suggesting that strategies with a higher contribution of grazed forage are detrimen-tal to mitochondrial function [13]. Few reports can be found studying protein acetylation in liver of dairy cows, however, a single study has shown that key enzymes related to fatty acid metabolism are differently acetylated in liver of cows with fatty liver disease and this could be involved with the pathogenesis of fatty liver disease [54]. In murine models, acetylation has been reported to inactivate mitochondrial enzymes related to fatty acid oxidation and deacety-lation is mediated by sirtuin 3 [55]. Studies in sirtuin 3 double knockout mice have indicated sirtuin 3 depletion increases hyperacetylation of mitochondrial proteins that are critical in the protection against hepatic lipotoxicity [56]. In particular, hyperacetylation of hepatic long-chain acyl coenzyme A dehydrogenase reduces its activity and this translates into a fatty acid oxidation disorder in sirtuin 3 double knockout mice which reduces ATP levels [57]. This murine model has also shown that ATP demanding pathways such as gluconeogenesis and the urea cycle may be affected due to reduced ATP availability [57] and failure to deacetylate key enzymes in the urea cycle [58].

## Conclusion

In this work we confirmed that hepatic mitochondrial function impairment of lactating dairy cows during early lactation is more aggravated in feeding strategies with higher proportions of grazed pasture. Besides, we found that CPT activity is also lower during early lactation. Both mitochondrial function and CPT activity are negatively correlated with liver triglyceride con-tent. Comparison of two different strains showed that North American Holstein (NAH) cows had the lowest values in respiratory parameters and highest levels of mitochondrial protein acetylation, in the strategy with higher proportions of grazed pasture (MaxP). While these parameters were not affected by diet in New Zealand Holstein (NZH) cows. These results sug-gest that increased grazed pasture inclusion in the diet is detrimental to hepatic energy

metabolism in the NAH strain. In addition, when hepatic gene expression was analyzed at 180 DIM, results showed that cows of the NZH strain had higher mRNA abundance of transcription factors and enzymes involved in mitochondrial and peroxisomal fatty acid oxidation. Overall, our results show that although cows of both Holstein strains mobilize lipid reserves–as measured by liver triglyceride levels–and have lower mitochondrial function and CPT activity during early lactation, NZH cows adapt their hepatic metabolism better to feeding strategies with greater proportions of grazed pasture (MaxP) as seen from lower levels of mitochondrial acetylation and an upregulation of oxidative metabolism in the liver. Our work highlights the relevance of selecting a suitable Holstein strain for pasture-based dairy systems.

## Supporting information

**S1 File. Western blots of mitochondrial acetylated lysine (AcK) and in mitochondrial fractions North American (NAH) and New Zealand (NZH) grazing Holstein cows with a fixed proportion of pasture (FixP) or maximum proportion of pasture (MaxP) at 21 and 180 DIM.** (N = 8). The last lane is acetylated BSA, bottom panel shows ponceau S staining for protein normalization. Western blots were quantified by densitometry, normalized with the loading control and expressed in relation to the average value of the NZH-FixP treatment at 21 DPP.
(PDF)

**S1 Table. Primers used for real time qPCR quantification.** [1]Genes: Very long-chain acyl-CoA dehydrogenase *(ACADVL)*, acetyl-CoA acetyltransferase 1 *(ACAT1)*, ß-actin *(ACTB)*, acyl-CoA oxidase 2 *(ACOX2)*, apolipoprotein A4 *(APOA4)*, apolipoprotein A5 *(APOA5)*, apolipoprotein C2 *(APOC2)*, CD36 molecule *(CD36)*, CD40 *molecule (CD40)*, carnitine palmitoyl-transferase 1 *(CPT1A)*, liver fatty acid binding protein *(FABP1)*, fibroblast growth factor 21 *(FGF21)*, hydroxymethylglutaryl-CoA synthase 2 *(HMGCS2)*, hypoxanthine phosphoribosyl transferase *(HPRT1)*; nuclear receptor subfamily 1 group H member 3 *(LXRA)*, nuclear factor kappa B subunit 1 *(NFKB1)*, nuclear factor kappa B inhibitor alpha *(NFKB1A)*, peroxisome proliferator-activated receptor alpha *(PPARA)*, peroxisome proliferator-activated receptor gamma coactivator 1-alpha *(PPARGC1A)*, retinoic acid receptor alpha *(RARA)*, retinoic X receptor alpha *(RXRA)*, retinoic X receptor beta *(RXRB)*, retinoic X receptor gamma *(RXRG)*, sterol regulatory element binding transcription factor 1 *(SREBP-1)*, tumor necrosis factor alpha *(TNFA)*, tumor necrosis factor receptor superfamily member 1A *(TNFRSF1A)*.
(DOCX)

**S2 Table. Hepatic complex-I respiration in liver biopsies of grazing Holstein cows of North American (NAH) and New Zealand (NZH) strains in two feeding strategies with varying proportions of grazed pasture during lactation.** Data are shown as least square means ± standard error. [ABab]Means having different uppercase superscripts differ significantly within feeding strategy across time period ($P < 0.05$). Means having different lowercase superscripts differ significantly within strain across time period ($P < 0.05$), while [XYxy]means denote tendencies ($0.05 < P < 0.1$). N = 10–12. DIM: Days in milk; FS: Feeding strategy. [1]Oxygen consumption rate measurements of liver biopsies were obtained after addition of 10 mM glutamate and 5 mM malate, 4 μM ADP, 2 μM oligomycin, up to 4 μM carbonyl cyanide-p- trifluoromethoxyphenylhydrazone and 0.5 μM rotenone, and 2.5 μM antimycin. Oxygen consumption rates are expressed as pmol of $O_2$ per min per mg of wet weight.
(DOCX)

**S3 Table. Hepatic complex-I respiration in liver biopsies of grazing Holstein cows of North American (NAH) and New Zealand (NZH) strains in two feeding strategies with**

**varying proportions of grazed pasture during lactation.** Data are shown as least square means ± standard error. [ABab]Means having different uppercase superscripts differ significantly within feeding strategy across time period (P < 0.05). Means having different lowercase superscripts differ significantly within strain across time period (P < 0.05), while [XYxy]means denote tendencies (0.05 < P < 0.1). N = 10–12. DIM: Days in milk; FS: Feeding strategy. [1]Oxygen consumption rate measurements of liver biopsies were obtained after addition of 20 mM succinate, 4 μM ADP, 2 μM oligomycin, up to 4 μM carbonyl cyanide-p- trifluoromethoxyphenyl-hydrazone and 0.5 μM rotenone, and 2.5 μM antimycin. Oxygen consumption rates are expressed as pmol of $O_2$ per min per mg of wet weight.
(DOCX)

**S4 Table. Hepatic specific CPT activity in liver biopsies of grazing Holstein cows of North American (NAH) and New Zealand (NZH) strains in two feeding strategies with varying proportions of grazed pasture during lactation.** Data are shown as least square means ± standard error. N = 10–12. DIM: Days in milk; FS: Feeding strategy.
(DOCX)

## Acknowledgments

The authors thank all the staff of the Dairy Unit of the Experimental Station "La Estanzuela" for their support in animal handling.

## Author Contributions

**Conceptualization:** Mercedes García-Roche, Alejandro Mendoza, Adriana Cassina, Celia Quijano, Mariana Carriquiry.

**Data curation:** Mariana Carriquiry.

**Formal analysis:** Mercedes García-Roche.

**Funding acquisition:** Alejandro Mendoza, Adriana Cassina, Celia Quijano, Mariana Carriquiry.

**Investigation:** Mercedes García-Roche, Daniel Talmón, Guillermo Cañibe, Ana Laura Astessiano, Alejandro Mendoza, Adriana Cassina, Celia Quijano, Mariana Carriquiry.

**Methodology:** Mercedes García-Roche, Daniel Talmón, Ana Laura Astessiano, Adriana Cassina, Celia Quijano, Mariana Carriquiry.

**Project administration:** Alejandro Mendoza, Adriana Cassina, Celia Quijano, Mariana Carriquiry.

**Resources:** Alejandro Mendoza, Adriana Cassina, Celia Quijano, Mariana Carriquiry.

**Supervision:** Alejandro Mendoza, Adriana Cassina, Celia Quijano, Mariana Carriquiry.

**Visualization:** Mercedes García-Roche, Guillermo Cañibe.

**Writing – original draft:** Mercedes García-Roche.

**Writing – review & editing:** Daniel Talmón, Guillermo Cañibe, Alejandro Mendoza, Celia Quijano, Mariana Carriquiry.

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
