## [Decision Letter · Decision Letter 0]

17 May 2023

PONE-D-23-08619Hepatic metabolism of grazing cows of two Holstein strains under two feeding strategies with different levels of pasture inclusionPLOS ONE

Dear Dr. Garcia Roche,

Thank you for submitting your manuscript to PLOS ONE. After careful consideration, we feel that it has merit but does not fully meet PLOS ONE’s publication criteria as it currently stands. Therefore, we invite you to submit a revised version of the manuscript that addresses the points raised during the review process.

The main problem is, liver metabolism changes need to be related to the difference in nutrient intake, digestive and metabolic rate. The discussion needs to be completely rewritten focusing on the interpretation of results relative to selection of Holstein strain for a particular management system or designing the management system for a particular Holstein strain. Thanks.

We look forward to receiving your revised manuscript.

Kind regards,

Arda Yildirim, Ph.D.

Academic Editor

PLOS ONE

Journal Requirements:

"Funding

M. García-Roche was supported by CAP fellowship BDDX_2018_1#49004502. D. Talmón was supported by ANII fellowship POS_NAC_2017_1_141266.  A. Cassina and C. Quijano were partially funded by grants of the Espacio Interdisciplinario – Centros, UDELAR 2015. A. Cassina was also supported by the grant CSIC grupos I+D 2014 (767). The project was funded by Comisión Sectorial de Investigación Científica (CSIC) of the Universidad de la República (UdelaR) CSIC I+D 2018 ID 103 to M. Carriquiry and C. Quijano as well as by Agencia Nacional de Investigación e Innovación (ANII) INNOVAGRO 2018: FSA_1_2018_1_152220 to M. Carriquiry and A. Cassina. A. Mendoza received funding from the project PL_21_0_00 of INIA."

3. Please expand the acronym “CAP, ANII, and INIA” (as indicated in your financial disclosure) so that it states the name of your funders in full.

6. Please clarify the Table 6 "Table 6. Inhibition by diphenyliodonium in liver biopsies of North American (NAH) and New Zealand (NZH) grazing Holstein cows supplemented with total mixed ration (FixP) or an energy-protein concentrate (MaxP) during prepartum and early lactation." in page "25" and Table 6 "

Table 6. Hepatic gene expression of North American (NAH) and New Zealand (NZH) grazing Holstein cows supplemented with total mixed ration (FixP) or an energy-protein concentrate (MaxP) at 180 DIM." in page "29".

Additional Editor Comments:

Dear Authors,

We have now received the required feedbacks from two reviewers and, before recommending your work for publishing, I think you should address the aspects they have highlighted, to further improve the quality of your manuscript. Thanks for your contribution to the Plos One. Best regards, Arda Yıldırım

Reviewers' comments:

Reviewer's Responses to Questions

**Comments to the Author**

1. Is the manuscript technically sound, and do the data support the conclusions?

Reviewer #1: Partly

Reviewer #2: Partly

2. Has the statistical analysis been performed appropriately and rigorously? 

Reviewer #1: Yes

Reviewer #2: Yes

3. Have the authors made all data underlying the findings in their manuscript fully available?

Reviewer #1: Yes

Reviewer #2: Yes

4. Is the manuscript presented in an intelligible fashion and written in standard English?

Reviewer #1: Yes

Reviewer #2: Yes

5. Review Comments to the Author

Reviewer #1: This manuscript describes an experiment evaluating hepatic metabolism of 2 Holstein strains in 2 feeding strategies of a grazing dairy at multiple time points during lactation. Overall, the experiment was performed well and data presentation is clear, but discussion of results is poor. The effect of treatments (3-way or 2-way interactions and main effects) on variables of hepatic metabolism are not consistent making interpretation very difficult. The current Discussion section does not help the reader interpret results at all. The Discussion section needs to be rewritten focusing on interpretation of results as to evaluating the appropriate strain for a particular feeding strategy or vice versa.

Specific comments:

L27 - please indicate that repeated measures was involved in this analysis

L30 - change to 'MaxP strategy at 21 DIM'

L31 - change to 'MaxP strategy at 21 DIM'

L56 - here and many other places in the manuscript you use a hyphen rather than a dash. The dash was correctly used in L55. please correct all occurrences.

L62 - change to 'associated with an increased'

L73 - change to 'associated with increased'

L105 - this is a 1 sentence paragraph

L111 and 112 - 'presented' should be 'represented'

L120 - is this referring to pasture intake, if so how did you measure that, or herbage allowance, if so how did you measure herbage mass

L123 - is 'paired' the block effect mentioned later in statistical analysis? If so you should use block term here

L126 - how do you know that pasture DMI was 33% of total DMI? did you measure pasture DMI?

L131 - what time of day were cows milked and how were they milked?

L133-134 - this sentence is confusing. what is annual average referring to?

L141 - how do you measure half a leaf

Table 1 - diet composition at -45 DIM is missing but the chemical composition is shown in Table 2

Table 2 - it is somewhat difficult to discern which columns fall under which headings

L186 - I assume since liver biopsies were cryopreserved that all mitochondrial respiration measurements for different DIM were performed at the same time. Measurements at different times would be affected by the consistency of the respiration medium making comparison of different DIM less relevant.

L230 - change to 'addition of the specific'

L240 - why were only 3 treatments measured for fatty acid driven respiration

L264 - change to 'presented in units'

L290 - are these the same 8 cows used for Western blots? Can PCR and western blot results be compared?

L314 - change to '(TNFA, and tumor'

L338 - is cow the correct experimental unit. I assume cows grazed as a group, but were they fed concentrate or TMR individually?

L343 - how many variables had more than 3 outliers

Table 3 - the P-value for strain x FS is missing

L394 - this is a 1 sentence paragraph

L399 - what is meant by moment of lactation? I am not familiar with that term and likely most readers are not either

Table 4 - how were glycogen/glucose ration and triglyceride/glycogen ratio calculated. my calculations based on data in the table do not match the values reported

L444 - change to 'interaction of strain'

L455 - change to 'between DIM and FS' to use consistent terminology

L468 - 'Alongside' seems unusual term here. suggest changing to 'Also'

Table 6 - why only -45 and 21 DIM. And why was FS not evaluated in this table

L524 - need to define AcK

L590 - why was CD36 only measured in NAH cows

L748 - change to 'content. North American'

L752 - change to 'metabolism in this strain.'

L748-756 - this explanation of the results is understandable and meaningful

Figure 2B - The interaction was not significant but looks like the means separation is indicating an interaction is present

Figure 3 - this seems to be the exact same data as in Table 5

Supplemental Table 1 and 2 - the superscripts seem to indicate there is a 3-way interaction but the p-value does not indicate an interaction. need to use different letters for superscripts to separate means of different main effects and interactions (abc, xyz, ABC, etc.)

Supplemental Table 1 - what is DIMPP

Reviewer #2: The manuscript characterized adaptations of hepatic metabolism of dairy cows of two Holstein strains with varying proportions of grazing in the feeding strategy.The design of the study is very interesting, the result confirmed the association between liver triglyceride, decreased hepatic mitochondrial function and greater mitochondrial acetylation levels in cows with a higher inclusion of pasture and suggests differential adaptative mechanisms between NAH and NZH cows to strategies with varying proportions of grazing in the feed strategy. The main problem is, liver metabolism changes need to be related to the difference of nutrient intake, digestive and metabolic rate.

Some specific comments:

1.L113. Please give us more details of the “economic and productive selection index”.

2.L121. How did you measure the intake of mixed pasture? Were the intakes of NZH, NAH cows designed at the same intake/body weight^0.75 ratio? If not, why the intake of NZH, NAH cows were designed at different level ?

3.Table 1. The total offered dry matter of NZH, NAH are different within the same stage. Intake level will affect the groth and metabolism of animals.

4.Table 1. Please calculate the main nutrient (crude protein, NDF, ADF, ME,et al.) intake.

5.Table 3. The a,b,c or x,y indicate that the data in the same row or line differ. Please specify whether the comparison is among the row or line. The superscripts are mislabeled in some rows, for example: milk yield at 21 DIM, superscript c did not appear, at 180 DIM, superscript a did not appear, and so on.Please check them carefully.

6.Fig.2. Please show us which one is NZH or NAH. The same as some other figures.

6. PLOS authors have the option to publish the peer review history of their article (what does this mean?). If published, this will include your full peer review and any attached files.

Reviewer #1: No

Reviewer #2: No

---

## [Author Response · Author response to Decision Letter 0]

3 Jul 2023

Journal Requirements:

"Funding

M. García-Roche was supported by CAP fellowship BDDX_2018_1#49004502. D. Talmón was supported by ANII fellowship POS_NAC_2017_1_141266. A. Cassina and C. Quijano were partially funded by grants of the Espacio Interdisciplinario – Centros, UDELAR 2015. A. Cassina was also supported by the grant CSIC grupos I+D 2014 (767). The project was funded by Comisión Sectorial de Investigación Científica (CSIC) of the Universidad de la República (UdelaR) CSIC I+D 2018 ID 103 to M. Carriquiry and C. Quijano as well as by Agencia Nacional de Investigación e Innovación (ANII) INNOVAGRO 2018: FSA_1_2018_1_152220 to M. Carriquiry and A. Cassina. A. Mendoza received funding from the project PL_21_0_00 of INIA."

AU: Sentence added.

3. Please expand the acronym “CAP, ANII, and INIA” (as indicated in your financial disclosure) so that it states the name of your funders in full.

AU: Thank you, acronyms were expanded.

AU: Added to supporting information (S5 file).

6. Please clarify the Table 6 "Table 6. Inhibition by diphenyliodonium in liver biopsies of North American (NAH) and New Zealand (NZH) grazing Holstein cows supplemented with total mixed ration (FixP) or an energy-protein concentrate (MaxP) during prepartum and early lactation." in page "25" and Table 6 "

Table 6. Hepatic gene expression of North American (NAH) and New Zealand (NZH) grazing Holstein cows supplemented with total mixed ration (FixP) or an energy-protein concentrate (MaxP) at 180 DIM." in page "29".

AU: Numbering has changed.

 AU: Captions included in the manuscript.

Additional Editor Comments:

Dear Authors,

We have now received the required feedbacks from two reviewers and, before recommending your work for publishing, I think you should address the aspects they have highlighted, to further improve the quality of your manuscript. Thanks for your contribution to the Plos One. Best regards, Arda Yıldırım

Reviewers' comments:

Reviewer's Responses to Questions

Comments to the Author

1. Is the manuscript technically sound, and do the data support the conclusions?

Reviewer #1: Partly

Reviewer #2: Partly

2. Has the statistical analysis been performed appropriately and rigorously?

Reviewer #1: Yes

Reviewer #2: Yes

3. Have the authors made all data underlying the findings in their manuscript fully available?

Reviewer #1: Yes

Reviewer #2: Yes

4. Is the manuscript presented in an intelligible fashion and written in standard English?

Reviewer #1: Yes

Reviewer #2: Yes

5. Review Comments to the Author

Thank you for your meaningful and constructive reviews. We have included more information and references to a publication of the same animal trial where animal and whole-farm biophysical performance were studied to support the Experimental Design section in Materials and Methods. In an effort to make the Results section more straight-forward, we avoided results which could translate into redundancies. For this, we excluded the free liver glucose to glycogen and liver glycogen to triglyceride ratios, non-significant correlations and the table that contained the strain x FS interaction for mitochondrial respiratory parameters as we found this information to be contained in other results. We also reanalyzed our results and modified the figures and tables to include all interactions. Figures were modified to include both two-way interactions: DIM x strain and DIM x FS and tables were modified to show only strain x FS means and all effects (fixed effects and two-way and three-way interactions)- Figures with the interaction strain x FS were included when found useful for the interpretation of results. In figure 5 we noticed that the graph with the correlation had incorrect x-axis legend, it was meant to be oligomycin-sensitive respiration, we included this correlation in the discussion section. Also, the order of the results was modified so it matched the order of the Discussion section. Finally, we modified the discussion by separating it into subsections with the headings: “Effect of strain on stage of lactation”, “Effect of feeding strategy on stage of lactation” and “Effect of feeding strategy and strain”.

Reviewer #1: This manuscript describes an experiment evaluating hepatic metabolism of 2 Holstein strains in 2 feeding strategies of a grazing dairy at multiple time points during lactation. Overall, the experiment was performed well and data presentation is clear, but discussion of results is poor. The effect of treatments (3-way or 2-way interactions and main effects) on variables of hepatic metabolism are not consistent making interpretation very difficult. The current Discussion section does not help the reader interpret results at all. The Discussion section needs to be rewritten focusing on interpretation of results as to evaluating the appropriate strain for a particular feeding strategy or vice versa.

Specific comments:

L27 - please indicate that repeated measures was involved in this analysis

AU: Thank you, it was added.

L30 - change to 'MaxP strategy at 21 DIM'

AU: Thank you, it was changed

L31 - change to 'MaxP strategy at 21 DIM'

AU: Thank you, it was changed

L56 - here and many other places in the manuscript you use a hyphen rather than a dash. The dash was correctly used in L55. please correct all occurrences.

AU: Thank you, checked.

L62 - change to 'associated with an increased'

AU: Thank you, it was changed

L73 - change to 'associated with increased'

AU: Thank you, it was changed

L105 - this is a 1 sentence paragraph

AU: Thank you, it was changed

L111 and 112 - 'presented' should be 'represented'

AU: Thank you, it was changed

L120 - is this referring to pasture intake, if so how did you measure that, or herbage allowance, if so how did you measure herbage mass

AU: Pre and post grazing pasture height was measured using a C-Dax Pasture Meter to estimate pasture intake, further detail was added to Materials and Methods. However, this information is present in a publication that specifically studied pasture growth and DM intake: - Stirling, S., L. Delaby, A. Mendoza, and S. Fariña. 2021. Intensification strategies for temperate hot-summer grazing dairy systems in South America: Effects of feeding strategy and cow genotype. J. Dairy Sci. 104 https://doi .org/10.3168/jds.2021-20507.

L123 - is 'paired' the block effect mentioned later in statistical analysis? If so you should use block term here

AU: Yes, it was changed for “blocked”.

L126 - how do you know that pasture DMI was 33% of total DMI? did you measure pasture DMI?

AU: Thank you, lines were added.

L131 - what time of day were cows milked and how were they milked?

AU: Milking time was added, cows were milked in a milking parlor with automated milk extraction, and automatic cluster removal. The latter information is not included since it is far from the objectives of this work, however the recording system was added. 

L133-134 - this sentence is confusing. what is annual average referring to?

AU: More information was added in Materials and Methods to clarify, 

L141 - how do you measure half a leaf

AU: It is in average two to three tillers or three tillers, however this information was excluded and a reference is provided to a publication that specifically studied pasture growth and DM intake : - Stirling, S., L. Delaby, A. Mendoza, and S. Fariña. 2021. Intensification strategies for temperate hot-summer grazing dairy systems in South America: Effects of feeding strategy and cow genotype. J. Dairy Sci. 104 https://doi .org/10.3168/jds.2021-20507.

Table 1 - diet composition at -45 DIM is missing but the chemical composition is shown in Table 2

AU: Prepartum diet was written in the text, it was now included to table 1.

Table 2 - it is somewhat difficult to discern which columns fall under which headings

AU: Thank you, lines were added.

L186 - I assume since liver biopsies were cryopreserved that all mitochondrial respiration measurements for different DIM were performed at the same time. Measurements at different times would be affected by the consistency of the respiration medium making comparison of different DIM less relevant.

AU: Thank you, we have clarified this observation in the Materials and Methods section. Lines 204-205

L230 - change to 'addition of the specific'

AU: Thank you, it was changed.

L240 - why were only 3 treatments measured for fatty acid driven respiration

AU: Thank you, we have clarified this observation in the Materials and Methods section. Lines 260-261

L264 - change to 'presented in units'

AU: Thank you, it was changed.

L290 - are these the same 8 cows used for Western blots? Can PCR and western blot results be compared?

AU: Thank you, we have clarified this observation in the Materials and Methods section. Lines 205-206

L314 - change to '(TNFA, and tumor'

AU: Thank you, it was changed.

L338 - is cow the correct experimental unit. I assume cows grazed as a group, but were they fed concentrate or TMR individually?

AU: Further detail is provided in the Materials and Methods section. The effect of strain is individual to every cow. With respect to the feeding strategies, the experiment consisted of a farmlet design that simulated two feedings strategies that reflect commercial practice (Stirling et al., 2021). In the MaxP strategy, cows were offered pasture depending on the pasture growth rate, with a heavy control on pasture growth. On the other hand, the FixP strategy had a moderate control on pasture growth and supplementation with TMR was offered on feedpads. To avoid dominance, cows grazed as a group within each strain and were all multiparous to avoid dominance. Also, cows were allocated to daily strips in a rotational manner to avoid any confounding effect from a given paddock. Concentrate was fed individually in the milking parlor to cows of the MaxP strategy, however, the TMR was offered in feedpads. In addition, the fact that most variables consisted of repeated measures minimizes the individual effect of the cow as previously mentioned by Bello et al., (2016).

L343 - how many variables had more than 3 outliers

AU: There were not variables with more than 3 outliers, variables with non-normal distribution were log-transformed. This observation was clarified in lines 360-362

Table 3 - the P-value for strain x FS is missing

AU: It was not included because it was not significant. It has now been included in the new table. 

L394 - this is a 1 sentence paragraph

L399 - what is meant by moment of lactation? I am not familiar with that term and likely most readers are not either

AU: Thank you, it was changed to stage of lactation.

Table 4 - how were glycogen/glucose ration and triglyceride/glycogen ratio calculated. my calculations based on data in the table do not match the values reported

AU: Thank you for your observation. Both ratios have a considerably high SEM, we have decided to exclude these variables from our results section. They do not provide additional information and may lead to misinterpretation of results. Also, we have made an effort to reduce the results section for a better interpretation of results. 

L444 - change to 'interaction of strain'

AU: Thank you, it was changed.

L455 - change to 'between DIM and FS' to use consistent terminology

AU: Thank you, it was changed.

L468 - 'Alongside' seems unusual term here. suggest changing to 'Also'

AU: Thank you, it was changed.

Table 6 - why only -45 and 21 DIM. And why was FS not evaluated in this table

AU: As we worked with cryopreserved samples and a great amount of samples we decided to prioritize this analysis for the transition period which is the period when dairy cows are most susceptible to metabolic disorders. Since no other interactions were significant we had not added this information, it has now been added. Lines 245-247.

L524 - need to define AcK

AU: Thank you, it was defined.

L590 - why was CD36 only measured in NAH cows

AU: All data on gene expression is provided in table 8, however graphs were performed only when P values were significant or tended to be significant to for better interpretation of results since over 20 genes were studied. Further detail is provided in the Materials and Methods section related to which results are presented in tables and which in graphs, lines 380-386.

L748 - change to 'content. North American'

AU: Thank you, it was changed.

L752 - change to 'metabolism in this strain.'

AU: Thank you, it was changed.

L748-756 - this explanation of the results is understandable and meaningful

AU: Thank you for your remark, we have modified the manuscript so it follows this rationale.

Figure 2B - The interaction was not significant but looks like the means separation is indicating an interaction is present

AU: Thank you, we agree with your observation, however it is the result we obtained from the statistical analysis. Bear in mind, the model includes the fixed effects of DIM, strain, FS and the corresponding two-way and three-way interactions. We have, nonetheless, excluded this figure in an effort to reduce the results section for a better interpretation of results. 

Figure 3 - this seems to be the exact same data as in Table 5

AU: Thank you, Table 5 was excluded, these results are visible in Figure 4 and Tables 1 and 2 of Supporting Information S2. 

Supplemental Table 1 and 2 - the superscripts seem to indicate there is a 3-way interaction but the p-value does not indicate an interaction. need to use different letters for superscripts to separate means of different main effects and interactions (abc, xyz, ABC, etc.)

AU: Thank you, it was changed

Supplemental Table 1 - what is DIMPP

AU: Thank you, it was changed

Reviewer #2: The manuscript characterized adaptations of hepatic metabolism of dairy cows of two Holstein strains with varying proportions of grazing in the feeding strategy.The design of the study is very interesting, the result confirmed the association between liver triglyceride, decreased hepatic mitochondrial function and greater mitochondrial acetylation levels in cows with a higher inclusion of pasture and suggests differential adaptative mechanisms between NAH and NZH cows to strategies with varying proportions of grazing in the feed strategy. The main problem is, liver metabolism changes need to be related to the difference of nutrient intake, digestive and metabolic rate.

Thank you for your meaningful and constructive reviews. We have included estimated nutrient intake in Materials and Methods and emphasized reduced dry matter intake in pasture-based systems in the Discussion section. 

Some specific comments:

1.L113. Please give us more details of the “economic and productive selection index”.

AU: Thank you, more information on the national economic-productive breeding value was included and two references were added (124-127 lines).

2.L121. How did you measure the intake of mixed pasture? Were the intakes of NZH, NAH cows designed at the same intake/body weight^0.75 ratio? If not, why the intake of NZH, NAH cows were designed at different level ?

AU: Pre and post grazing pasture height was measured using a C-Dax Pasture Meter to estimate pasture intake, further detail was added to Materials and Methods (150-157 lines). Predicted DM intake was estimated with NRC 2001 which takes into account fat corrected milk, metabolic BW and week of lactation. This information is present in a publication that specifically studied pasture growth and DM intake: - Stirling, S., L. Delaby, A. Mendoza, and S. Fariña. 2021. Intensification strategies for temperate hot-summer grazing dairy systems in South America: Effects of feeding strategy and cow genotype. J. Dairy Sci. 104 https://doi .org/10.3168/jds.2021-20507. 

3.Table 1. The total offered dry matter of NZH, NAH are different within the same stage. Intake level will affect the groth and metabolism of animals.

AU: Thank you for the remark, table 1 was incorrectly named, it is the estimated dry matter intake according to pre and post grazing pasture height and TMR and concentrate offered and refusals. Pasture allowance was allocated according to pasture growth rate and predicted DM intake. Further detail has been included in the Materials and Methods section for clarification.

We agree intake level may affect growth and metabolism of animals, as mentioned in the Introduction (66-71 lines) and Discussion (733-737 and 781-790 lines) intake is limited in pasture-based systems and becomes more limited with increasing proportions of grazed pasture. 

4.Table 1. Please calculate the main nutrient (crude protein, NDF, ADF, ME,et al.) intake.

AU: Thank you, it was included in table 1. 

5.Table 3. The a,b,c or x,y indicate that the data in the same row or line differ. Please specify whether the comparison is among the row or line. The superscripts are mislabeled in some rows, for example: milk yield at 21 DIM, superscript c did not appear, at 180 DIM, superscript a did not appear, and so on.Please check them carefully.

AU: Comparisons are not within the same row, they are within the same variable, letters c and d in milk yield do not appear in 21 DIM because milk yield is lower at 180 DIM, vice versa for letter a in 180 DIM. 

However, to clarify the interpretation of results, we have included two panels: one for DIM x Strain interaction and another for DIM x FS interaction while the Strain x FS interaction is presented mostly in tables, except for two figures to aid with the discussion of results. 

6.Fig.2. Please show us which one is NZH or NAH. The same as some other figures.

AU: To clarify the interpretation of results, the results section was modified in order to show DIM x FS in one panel and DIM x Strain in the other, the Strain x FS interaction is shown in tables. The DIM x Strain interaction was not shown in graphs since it was not significant in most variables. Given that the three-way interaction was not significant for most of the results (only %protein in milk) it was not graphed so as to make interpretation of results more simple.

6. PLOS authors have the option to publish the peer review history of their article (what does this mean?). If published, this will include your full peer review and any attached files.

Do you want your identity to be public for this peer review? For information about this choice, including consent withdrawal, please see our Privacy Policy.

Reviewer #1: No

Reviewer #2: No

---

## [Decision Letter · Decision Letter 1]

10 Aug 2023

Hepatic metabolism of grazing cows of two Holstein strains under two feeding strategies with different levels of pasture inclusion

PONE-D-23-08619R1

Dear Dr. Garcia Roche,

We’re pleased to inform you that your manuscript has been judged scientifically suitable for publication and will be formally accepted for publication once it meets all outstanding technical requirements.

Kind regards,

Arda Yildirim, Ph.D.

Academic Editor

PLOS ONE

Additional Editor Comments (optional):

Dear Authors, Your exceptional work and the revisions you made in response to the reviewers' feedback have greatly strengthened the manuscript. Thank you for your valuable work and for choosing Plos One as the platform to share your research. We look forward to the publication and the positive impact it will have. Regards, Arda Yıldırım.

Reviewers' comments:

Reviewer's Responses to Questions

**Comments to the Author**

1. If the authors have adequately addressed your comments raised in a previous round of review and you feel that this manuscript is now acceptable for publication, you may indicate that here to bypass the “Comments to the Author” section, enter your conflict of interest statement in the “Confidential to Editor” section, and submit your "Accept" recommendation.

Reviewer #1: All comments have been addressed

Reviewer #2: All comments have been addressed

2. Is the manuscript technically sound, and do the data support the conclusions?

Reviewer #1: Yes

Reviewer #2: (No Response)

3. Has the statistical analysis been performed appropriately and rigorously? 

Reviewer #1: Yes

Reviewer #2: (No Response)

4. Have the authors made all data underlying the findings in their manuscript fully available?

Reviewer #1: Yes

Reviewer #2: (No Response)

5. Is the manuscript presented in an intelligible fashion and written in standard English?

Reviewer #1: Yes

Reviewer #2: (No Response)

6. Review Comments to the Author

Reviewer #1: (No Response)

Reviewer #2: (No Response)

7. PLOS authors have the option to publish the peer review history of their article (what does this mean?). If published, this will include your full peer review and any attached files.

Reviewer #1: No

Reviewer #2: No

---

## [Editor Report · Acceptance letter]

17 Aug 2023

PONE-D-23-08619R1 

Hepatic metabolism of grazing cows of two Holstein strains under two feeding strategies with different levels of pasture inclusion 

Dear Dr. Garcia Roche:

I'm pleased to inform you that your manuscript has been deemed suitable for publication in PLOS ONE. Congratulations! Your manuscript is now with our production department. 

Kind regards, 

on behalf of

Prof. Dr. Arda Yildirim 

Academic Editor

PLOS ONE